# A digital 3D reference atlas reveals cellular growth patterns shaping the *Arabidopsis* ovule

Athul Vijayan[1†], Rachele Tofanelli[1†], Sören Strauss[2], Lorenzo Cerrone[3], Adrian Wolny[3,4], Joanna Strohmeier[1], Anna Kreshuk[4], Fred A Hamprecht[3], Richard S Smith[2‡], Kay Schneitz[1*]

[1]Plant Developmental Biology, School of Life Sciences, Technical University of Munich, Freising, Germany; [2]Department of Comparative Development and Genetics, Max Planck Institute for Plant Breeding Research, Cologne, Germany; [3]Heidelberg Collaboratory for Image Processing, Dept. of Physics and Astronomy, Heidelberg University, Heidelberg, Germany; [4]European Molecular Biology Laboratory, Heidelberg, Germany

**Abstract** A fundamental question in biology is how morphogenesis integrates the multitude of processes that act at different scales, ranging from the molecular control of gene expression to cellular coordination in a tissue. Using machine-learning-based digital image analysis, we generated a three-dimensional atlas of ovule development in *Arabidopsis thaliana*, enabling the quantitative spatio-temporal analysis of cellular and gene expression patterns with cell and tissue resolution. We discovered novel morphological manifestations of ovule polarity, a new mode of cell layer formation, and previously unrecognized subepidermal cell populations that initiate ovule curvature. The data suggest an irregular cellular build-up of *WUSCHEL* expression in the primordium and new functions for *INNER NO OUTER* in restricting nucellar cell proliferation and the organization of the interior chalaza. Our work demonstrates the analytical power of a three-dimensional digital representation when studying the morphogenesis of an organ of complex architecture that eventually consists of 1900 cells.

**\*For correspondence:**
kay.schneitz@tum.de

[†]These authors contributed equally to this work

**Present address:** [‡]The John Innes Centre, Norwich Research Park, Norwich, United Kingdom

**Competing interests:** The authors declare that no competing interests exist.

## Introduction

How organs attain their species-specific size and shape in a reproducible manner is an important question in biology. Tissue morphogenesis constitutes a multi-scale process that occurs in three dimensions (3D). Thus, quantitative cell and developmental biology must address not only molecular processes but also cellular and tissue-level properties. It necessitates the quantitative 3D analysis of cell size, cell shape, and cellular topology, an approach that has received much less attention (*Boutros et al., 2015*; *Jackson et al., 2019*).

Morphogenesis involves the coordination of cellular behavior between cells or complex populations of cells, leading to emergent properties of the tissue that are not directly encoded in the genome (*Coen et al., 2017*; *Coen and Rebocho, 2016*; *Gibson et al., 2011*; *Jackson et al., 2019*). For example, plant cells are linked through their cell walls. The physical coupling of plant cells can cause mechanical stresses that may control tissue shape by influencing growth patterns and gene expression (*Bassel et al., 2014*; *Hamant et al., 2008*; *Hervieux et al., 2016*; *Kierzkowski et al., 2012*; *Landrein et al., 2015*; *Louveaux et al., 2016*; *Sampathkumar et al., 2014*; *Sapala et al., 2018*; *Sassi et al., 2014*; *Uyttewaal et al., 2012*). Concepts involving the minimization of mechanical stresses caused by differential growth within a tissue have been employed to explain

morphogenesis of different plant organs with curved shapes (*Lee et al., 2019*; *Liang and Mahadevan, 2011*; *Rebocho et al., 2017*).

Developmental changes in the appearance of organs, such as leaves and sepals, are often assessed by focusing on the organ surface (*Hervieux et al., 2016*; *Hong et al., 2016*; *Kierzkowski et al., 2019*). This strategy, however, neglects internal cellular growth patterns. Cellular patterns in deeper tissue layers have classically been studied using 2D sectioning techniques with modern variations, for example in the study of the hypocotyl, relying on automated quantitative histology (*Sankar et al., 2014*). However, 2D analysis of cellular patterns can also result in misconceptions as was noticed for the early Arabidopsis embryo (*Yoshida et al., 2014*).

For a more complete understanding of morphogenesis, a 3D cellular level description and quantification of the entire tissue under study is essential (*Hong et al., 2018*; *Kierzkowski and Routier-Kierzkowska, 2019*; *Sapala et al., 2019*). Digital 3D organs with cellular resolution are a desired goal, however, it remains a challenge to generate such accurate digital representations. Substantial efforts in animal developmental biology often still lack single-cell resolution (*Anderson et al., 2019*; *Asadulina et al., 2012*; *Dreyer et al., 2010*; *Lein et al., 2007*; *Rein et al., 2002*). Notable exceptions are *Caenorhabditis elegans* (*Long et al., 2009*) and the early embryo of the ascidian *Phallusia mammillata* (*Guignard et al., 2020*; *Sladitschek et al., 2020*). Model plants, such as *Arabidopsis thaliana*, are uniquely suited for this task. Plants feature a relatively small number of different cell types. Moreover, plant cells are immobile. As a consequence, one can often observe characteristic cell division patterns associated with the formation and organization of tissues and organs. A cellular level 3D digital organ atlas has been obtained for the early stages of the Arabidopsis embryo (*Yoshida et al., 2014*) which has very few cells. Mostly, complete atlases have been made for larger organs with simple layered structures like roots and hypocotyls (*Bassel et al., 2014*; *Montenegro-Johnson et al., 2015*; *Pasternak et al., 2017*; *Schmidt et al., 2014*; *Yoshida et al., 2014*). The applied approach, however, was incompatible with fluorescent stains. Full 3D processing of live imaged data sets has also been possible in some systems, such as the shoot apex but is limited by light penetration to outer layers (*Montenegro-Johnson et al., 2019*; *Refahi et al., 2020*; *Willis et al., 2016*).

The ovule is the female reproductive organ of higher plants. It harbors the embryo sac with the egg cell that is protected by two integuments, lateral determinate tissues that develop into the seed coat following fertilization. The Arabidopsis ovule has been established as a model to study several important aspects of tissue morphogenesis including primordium formation, the establishment of the female germ line, and integument formation (*Chaudhary et al., 2018*; *Gasser and Skinner, 2019*; *Nakajima, 2018*; *Schmidt et al., 2015*). The ovule exhibits an elaborate tissue architecture exemplified by its multiple cell types and extreme curvature visible in the highly asymmetric growth of the two integuments. This property makes it ideal for addressing the complexity of morphogenetic processes. Qualitative descriptions of ovule development exist (*Christensen et al., 1997*; *Robinson-Beers et al., 1992*; *Schneitz et al., 1995*) but a quantitative cellular characterization is only available for the tissue that forms the germ line (*Lora et al., 2017*; *Hernandez-Lagana et al., 2020*).

To make the next step in the study of ovule morphogenesis therefore requires 3D digital ovules with cellular resolution over all developmental stages. Here, we constructed a canonical 3D digital atlas of Arabidopsis ovule development with cellular resolution. The atlas covers all stages from early primordium outgrowth to the mature pre-fertilization ovule and provides quantitative information about various cellular parameters. It also provides a proof-of-concept analysis of spatial gene expression in 3D with cellular resolution. Our quantitative phenotypic analysis revealed a range of novel aspects of ovule morphogenesis and a new function for the regulatory gene *INNER NO OUTER*.

## Results

Arabidopsis ovules become apparent as finger-like protrusions that emanate from the placental tissue of the carpel (*Robinson-Beers et al., 1992*; *Schneitz et al., 1995*; *Figure 1A*). The ovule is a composite of three clonally distinct radial layers (*Jenik and Irish, 2000*; *Schneitz et al., 1995*). Thus, its organization into L1 (epidermis), L2 (first subepidermal layer), and L3 (innermost layer) follows a general principle of plant organ architecture (*Satina et al., 1940*). Following primordium formation, three proximal-distal (PD) pattern elements can be recognized: the distal nucellus, central chalaza, and proximal funiculus, respectively (*Figure 1A,B*). The nucellus produces the megaspore mother

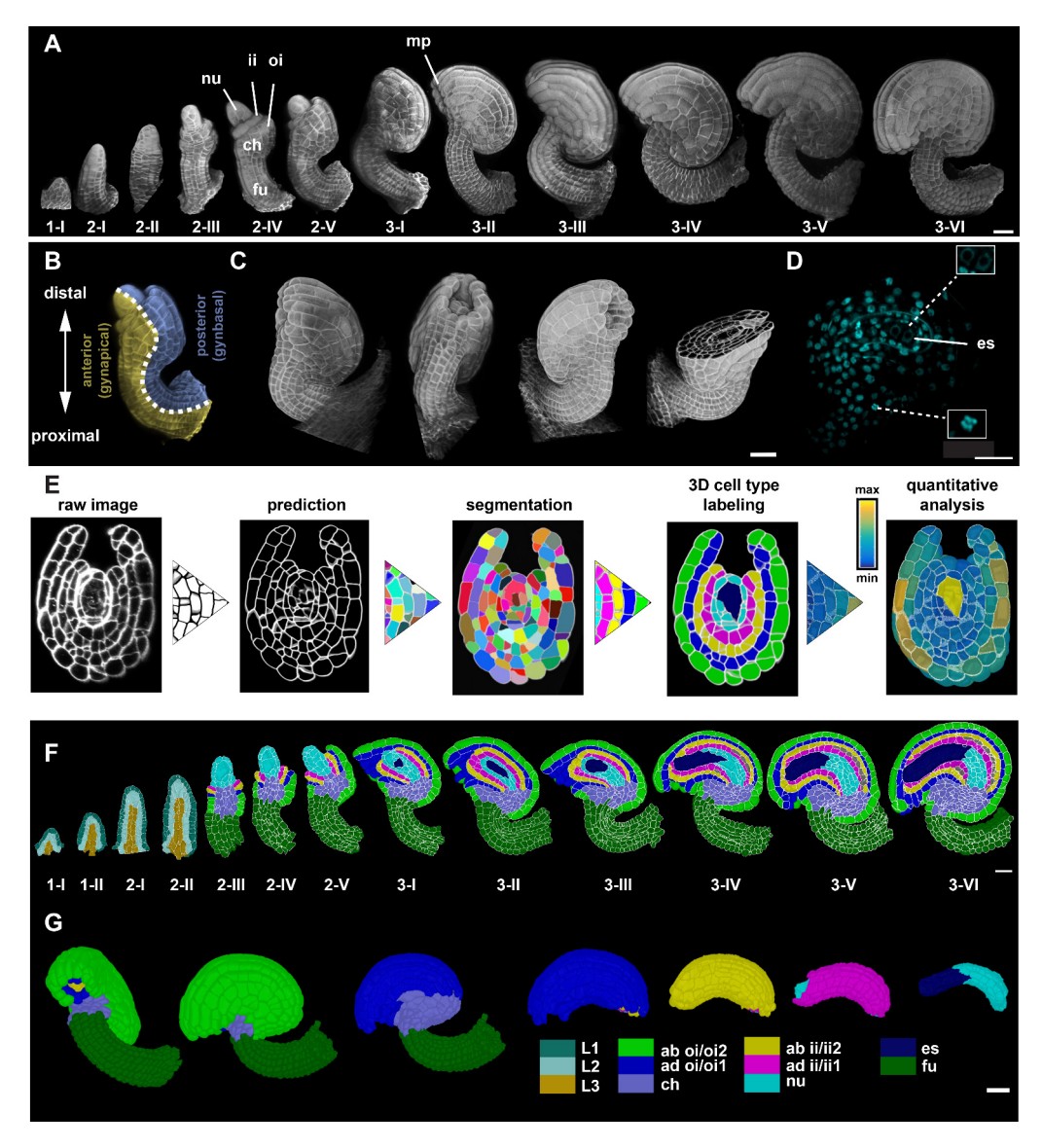

**Figure 1.** Stage-specific 3D digital ovules with cellular and tissue resolution. (**A**) 3D rendering of confocal z-stacks of SR2200-stained cell walls of ovule depicting ovule development from initiation at stage 1-I to maturity at stage 3-VI. (**B**) The different polarities of the ovule: the proximal-distal axis and the anterior-posterior (gynapical-gynbasal) axis are indicated. (**C**) 3D rendering of confocal z-stacks with multi-view of an ovule depicting the quality of the raw microscopic image. (**D**) Mid-section clip plane from the TO-PRO-3 channel displaying a two-nuclear embryo sac and mitotic nuclei. (**E**) Pipeline generating 3D digital ovules: raw data, PlantSeg cell contour prediction, 3D GASP segmentation, cell type annotation and quantitative analysis. (**F**) Mid-sagittal section of ovules from stages 1-I to 3-VI showing the cell type organization in wild-type ovules. Stages 1-I to 2-II includes radial L1, L2, L3 labeling. From stage 2-III, individual cell type labels are assigned according to the specific tissue. (**G**) 3D view of a mature ovule with cell type labels. The inner tissues are extracted from the 3D ovule after removing the overlying tissues and visualized separately. Different colors represent different tissue type labels. ii1/ ii2, oi1/oi2 designate the integument layers as described in *Beeckman et al., 2000*. Number of 3D digital ovules scored: 10 (stages 2-III, 2-IV, 2-V, 3-I, 3-II, 3-IV, 3-VI), 11 (3-III, 3-V), 13 (stage 2-II), 23 (stage 1-I), 49 (stage 2-I), 66 (stage 1-II). Abbreviations: ab, abaxial; ad, adaxial; ch, chalaza; es, embryo sac; fu, funiculus; ii, inner integument; mp, micropyle; nu, nucellus; oi, outer integument. Scale bars: 20 µm.

The online version of this article includes the following source data for figure 1:

**Source data 1.** Includes information of the available wild-type dataset of ovules at different developmental stages and their respective IDs.

cell (MMC), a large L2-derived cell that eventually undergoes meiosis. Only one of the meiotic products, the functional megaspore, survives and continues development. It develops into the eight-nuclear, seven-celled haploid embryo sac. The embryo sac, or female gametophyte, carries the egg cell. The chalaza is characterized by two integuments that initiate at its flanks. The two sheet-like integuments are determinate lateral organs of epidermal origin that undergo planar or laminar growth (*Jenik and Irish, 2000*; *Schneitz et al., 1995*; *Truernit and Haseloff, 2008*). The integuments grow around the nucellus in an asymmetric fashion eventually forming a hood-like structure and contributing to the curved shape (anatropy) of the mature ovule. Each of the two integuments initially forms a bilayered structure of regularly arranged cells. Eventually, the inner integument forms a third layer. The outer integument consists of two cell layers throughout its development. The two integuments leave open a small cleft, the micropyle, through which a pollen tube can reach the interior of the ovule (*Figure 1A,C*). The funiculus represents a stalk-like structure that carries the vasculature and connects the ovule to the placenta. Ovules eventually orient along the apical-basal or long axis of the gynoecium (pistil) with the micropyle facing toward the stigma (anterior half of ovule oriented gynapically), whereas the opposite side of the ovule faces the bottom of the gynoecium (posterior half, oriented gynbasally) (*Simon et al., 2012*; *Figure 1B*, see below).

## Generating stage-specific 3D digital ovules with cellular resolution

Ovules are buried within the gynoecium and live imaging of Arabidopsis ovule development is not feasible, except for a short period of time and with a focus on a given cell (*Tofanelli et al., 2019*; *Valuchova et al., 2020*). Thus, we resorted to imaging cohorts of fixed specimens. We obtained z-stacks of optical sections of fixed and cleared ovules at different stages by laser scanning confocal microscopy (CLSM). The image stacks were further handled using `MorphoGraphX` (MGX) software (*Barbier de Reuille et al., 2015*; *Strauss et al., 2019*). The imaging method has recently been described in detail (*Tofanelli et al., 2019*). In short, we dissected and fixed ovules of different stages, cleared the ovules using ClearSee (*Kurihara et al., 2015*) and simultaneously stained the cell wall and nuclei using the fluorescent stains SR2200 (*Musielak et al., 2015*) and TO-PRO-3 iodide (TO-PRO-3) (*Bink et al., 2001*; *Van Hooijdonk et al., 1994*), respectively (*Figure 1C,D*).

We processed the raw 3D datasets using PlantSeg, a deep learning pipeline for 3D segmentation of dense plant tissue at cellular resolution (*Wolny et al., 2020*; *Figure 1E*) (Materials and methods). The pipeline includes two major steps: cell wall stain-based cell boundary prediction performed by a convolutional neural network (CNN) and 3D cell segmentation based on the respective cell boundary predictions. Even after extensive optimizations, the procedure still resulted in some mistakes. We found that two distinct groups of cells represented the main sources of error. The first group included the MMC and its direct lateral neighbors at stages 2-III to 2-V. The second group encompassed the cells of the late embryo sac (stages 3-V/3-VI) (*Tofanelli et al., 2019*). We believe the reason for these errors lies in poor staining with SR2200 and could be due to their cell walls being particularly thin or of a biochemical composition recalcitrant to staining. Poor staining of the embryo sac cells could be explained by the observation that they exhibit only partially formed cell walls (*Mansfield et al., 1991*). An additional minor source of errors related to remaining general segmentation mistakes that were partly manually corrected. For further analysis, we selected ovules devoid of apparent segmentation errors for stages 1 to 2-II and 3-I to 3-IV. For stages 2-III to 2-V, we included ovules containing no more than five under-segmented (uncorrected) cells in the region occupied by the MMC and its lateral neighboring cells ($\leq$10% of nucellar cells). Regarding mature ovules (stages 3-V/3-VI) we included ovules devoid of apparent segmentation errors in the sporophytic tissue.

## Cell-type labeling of 3D digital ovules with cellular resolution

Following the generation of 3D cell meshes, we added specific labels to individual cells, thereby describing tissue types, such as radial cell layers (L1, L2, L3), nucellus, internal tissue of the chalaza, inner or outer integument, or the funiculus (see *Supplementary file 1* for cell types). Staining with TO-PRO-3 also allowed the identification of cells undergoing mitosis (*Figure 1D*). Available computational pipelines for near-automatic, geometry-based cell type identification (*Montenegro-Johnson et al., 2015*; *Montenegro-Johnson et al., 2019*; *Schmidt et al., 2014*) failed to provide reasonably good and consistent results. This was likely due to the ovule exhibiting a more complex

tissue architecture. We therefore performed cell type labeling by a combination of semi-automated and manual cell type labeling. The entire procedure, from imaging to the final segmented and labeled digital ovule, takes about 45 min per z-stack. For younger ovules, the procedure takes even less time.

In summary, the combined efforts resulted in a high-quality reference set of 158 hand-curated 3D digital ovules of the wild-type Col-0 accession (≥10 samples per stage). They feature cellular resolution, cover all stages, and include annotated cell types and cellular features (*Figure 1F*; *Video 1*).

## Overall assessment of ovule development

The different stages of Arabidopsis ovule development were defined previously (*Schneitz et al., 1995*). Throughout this work, we used a more precise definition of the subdivision of stage 1 into stage 1-I and 1-II. The subdivision was based on the first appearance of the signal of a reporter for *WUSCHEL* expression that became robustly apparent when ovule primordia consisted of 50 cells (see below). We first determined the average number of cells per ovule and stage (*Figure 2A,B*) and found an incremental increase in cell number for every consecutive stage of development until ovules at stage 3-VI exhibited an average of about 1900 cells (1897 ± 179.9 (mean ± SD)) (*Table 1*). We also assessed the mean volume per ovule for each stage by summing up the cell volumes of all cells in a given ovule (*Figure 2A,C*; *Table 1*). We measured a mean total volume for ovules at stage 3-VI of about $5 \times 10^5$ µm$^3$ ($4.9 \times 10^5 \pm 0.7 \times 10^5$).

## Relative tissue growth patterns and dynamics during ovule development

To get a first insight into potential tissue-specific growth patterns, we performed a stage-specific, quantitative cellular analysis of different tissues following primordium formation (*Table 2*). We observed little if any growth of the nucellus (excluding the embryo sac) except at the end of stage 3, with the funiculus halting growth at stage 3-II, while the cell numbers and tissue volume of the chalaza and integuments continuously increased during stages 2 and 3 (for a detailed description see Appendix1 with *Appendix 1—figure 1* and *Appendix 1—figure 2*).

To gain more insight into the growth dynamics during ovule development, we assessed its temporal progression. To this end, we established a two-step procedure. First, we investigated pistil growth for the covered developmental period by measuring pistil length of individual live pistils attached to the plant at different times spanning floral stages 8–12 (stages according to *Smyth et al., 1990*; *Figure 3A*). We then generated a pistil growth curve by fitting a curve to a plot of pistil length over time (*Figure 3B*) (*Supplementary file 2*). Next, we dissected ovules from pistils of various lengths, determined the stage of the ovules for each pistil, and used the pistil growth curve for an estimate of the duration of the different ovule stages (see Materials and methods). It was previously noticed that early stage ovules do not develop synchronously within a pistil (*Schneitz et al., 1995*; *Wolny et al., 2020*; *Yu et al., 2020*). In line with these results, we observed ovules of different stages within a given pistil (*Supplementary file 3*). Asynchrony was obvious for all stages and ovules within a pistil were usually found to be distributed across two to four consecutive stages. We determined the number of ovules per stage for a given pistil and defined the stage with the largest number of ovules as reference. Stages 3-II and 3-III were grouped together as pistils usually contained about equal numbers of both stages. From the pistil growth curve, we then derived the time interval between the minimal and maximal pistil lengths covering a given stage. This time interval

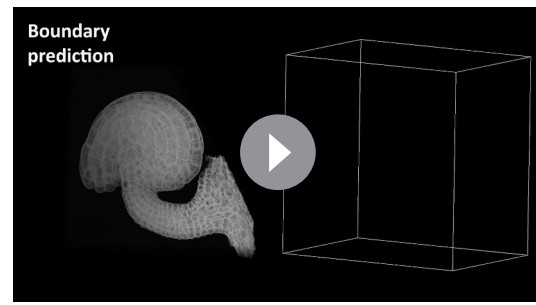

**Video 1.** Mature 3D digital ovule with cell and tissue resolution. Video represents the summary of the workflow and multidimensional view of a 3D digital ovule of stage 3-IV generated from a z stack of cell wall images. Different colors represent 3D cells grouped according to respective tissue type labels. To the end, the 3D surfaces of inner tissues are extracted from the 3D ovule after removing the overlying tissues and visualized separately.

https://elifesciences.org/articles/63262#video1

provided an estimate of the duration of a given ovule stage (*Figure 3C*).

Under our growth conditions, we observed that ovules emerge from placental tissue in pistils with a length of about 0.33 mm and that by the time of anthesis pistils had reached a length of about 2.3 mm. Ovules developed from early stage 1 to the end of stage 3 within a period of 146.2 hr or 6.1 days. The three main stages differed in their durations. Stage 1 required about 31.7 hr (1.32 days), while stage 2 was 58.9 hr (2.45 days) and stage 3 was 55.6 hr (2.32 days), roughly 1.9 or 1.8 times longer, respectively. These results provide a refined timeline for ovule development when compared to previous estimates (*Schneitz et al., 1995*). Using the mean ovule volume per stage (*Table 1*), we estimated average growth rates for certain time intervals. For up to the end of stage 1-II, we estimated an average growth rate of $0.3 \times 10^3$ µm$^3$/hr, of $1.5 \times 10^3$ µm$^3$/hr for the interval including stages 2-I to , and of $7.2 \times 10^3$ µm$^3$/hr for the interval including stages 3-I to 3-IV. The growth rate dropped to $2.9 \times 10^3$ µm$^3$/hr for stage 3-V.

We then investigated relative growth (*Figure 4*). In a first step, we normalized global growth and cell proliferation rates during two consecutive stages by the respective rate of the stage 1-I/1-II interval (*Figure 4A*). Both relative growth rate curves showed a dynamic behavior across ovule development and for several stage intervals we determined similar relative growth rates. Interestingly, however, we noticed that from stages 2-I to 2-II the relative cell proliferation rate was higher than the relative growth rate suggesting growth accompanied by cell proliferation. By contrast, the opposite was observed for stages 2-IV to 3-IV indicating that ovule growth during these stages was mainly due to cell expansion. In a second step, we investigated the relative growth of the ovule by plotting the ratio between the mean total volume or mean total number of cells for two consecutive stages (*Figure 4B*). The curves in *Figure 4A and B* qualitatively resemble each other. This is presumably due to the relatively minor variations in the duration of the individual ovule stages.

Next, we investigated the relative growth of the major tissues (*Figure 4C–J*). We took the ratio between the mean total volume or mean total cell number of a tissue for two consecutive stages and divided it by the corresponding relative ovule growth. The ratio relates the tissue growth rate to overall ovule growth. Values above one indicate that the tissue is growing at a higher rate than overall ovule growth rate and values below one indicate the opposite. We observed that the L3 showed the most relative growth and the L1 the least during stage 1-I (*Figure 4C–F*). During stages 1-II to 2-II, all three layers contributed similarly to the overall growth. From stages 2-III to 3-VI, the nucellus, chalaza, integuments, and eventually the embryo sac showed dynamic changes during development (*Figure 4G–J*). For example, up to stage 3-I the outer integument exhibited more relative growth than the inner integument, while from stage 3-I on the relative growth pattern was reversed.

In summary, the results reveal ovule growth to be dynamic both in terms of overall growth and the respective relative contributions of individual tissues during development.

## Scattered buildup of *WUSCHEL* expression in the nucellus

3D digital ovules allow a detailed investigation of spatial gene expression patterns and with cellular resolution. As proof-of-concept, we analyzed the expression of *WUSCHEL* (*WUS*, AT2G17950) in the young ovule. *WUS* controls stem cell development in the shoot apical meristem (*Gaillochet et al., 2015*; *Mayer et al., 1998*). *WUS* is also active during ovule development and promotes early pattern formation in the chalaza and integument initiation as *wus* mutants show misorganized gene expression in the chalaza and lack integuments (*Gross-Hardt et al., 2002*; *Sieber et al., 2004*). Moreover, *WUS* is also required for MMC formation (*Lieber et al., 2011*) but its expression must be excluded from the MMC to allow proper progress through meiosis (*Zhao et al., 2017*). A combination of in situ hybridization experiments, reporter gene analyses, and immunodetection indicated that *WUS* is expressed in the epidermis of the nucellus. Thus, it is currently thought that *WUS* performs its functions in a non-cell-autonomous fashion (*Gross-Hardt et al., 2002*; *Lieber et al., 2011*; *Sieber et al., 2004*; *Zhao et al., 2017*). However, it remains unclear when *WUS* expression first appears and thus at what stage it may become functionally relevant, whether its expression is always restricted to the nucellus, and whether *WUS* expression is limited to the nucellar epidermis or is present in L2 cells (and possibly the MMC) as well. To address these issues, we took advantage of a Col reporter line carrying a reporter for *WUS* promoter activity (pWUS::2xVENUS:NLS::tWUS (pWUS)) (*Zhao et al., 2017*). We generated a total of 67 3D digital ovules from the pWUS reporter line covering stages 1-I to 2-III ($9 \leq n \leq 21$ per stage).

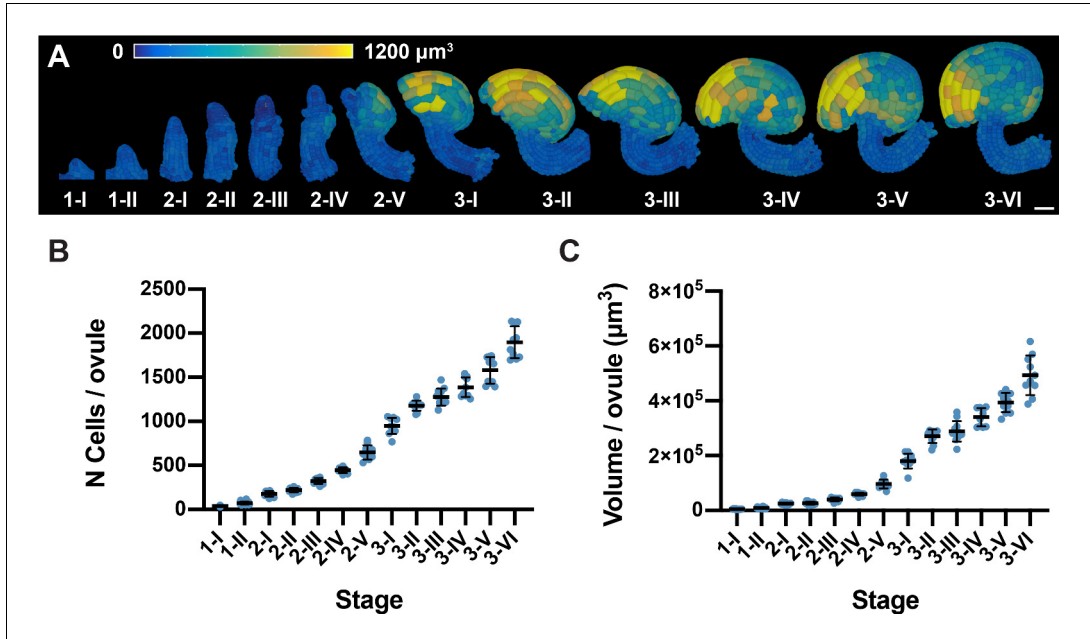

**Figure 2.** Ovule developmental stages and overall growth patterns. (**A**) 3D cell mesh view of wild-type ovules at different stages displaying heatmaps of cell volume ranging from 0 to 1200 μm³. (**B, C**) Plots depicting the total number of cells and total volume of individual ovules from early to late stages of development, respectively. Number of 3D digital ovules scored: 10 (stages 2-III, 2-IV, 2-V, 3-I, 3-II, 3-IV, 3-VI), 11 (3-III, 3-V), 13 (stage 2-II), 23 (stage 1-I), 49 (stage 2-I), 66 (stage 1-II). Mean ± SD is shown. Scale bar: 20 μm.

The online version of this article includes the following source data for figure 2:

**Source data 1.** Includes the list of ovule IDs, stage, total number of cells and total volume of the available wild-type dataset.

We first asked when the pWUS signal became detectable during ovule development. We found that with two exceptions pWUS signals could robustly be observed starting with ovules carrying 50

**Table 1.** Cell numbers and total volumes of ovules at different stages.

| Stage* | N cells | Volume (x10⁴ μm³) | N mitotic cells | % mitotic cells |
|---|---|---|---|---|
| 1-I | 39.6 ± 5.3 | 0.5 ± 0.09 | 1.0 ± 0.0 | 0.7 ± 1.2 |
| 1-II | 74.0 ± 17.1 | 1.0 ± 0.2 | 1.3 ± 0.5 | 0.7 ± 0.9 |
| 2-I | 176.9 ± 31.5 | 2.5 ± 0.4 | 3.1 ± 2.1 | 1.8 ± 1.2 |
| 2-II | 220.6 ± 24.9 | 2.7 ± 0.6 | 2.7 ± 1.6 | 1.1 ± 0.7 |
| 2-III | 324.1 ± 32.9 | 4.1 ± 0.7 | 3.6 ± 1.7 | 1.0 ± 0.7 |
| 2-IV | 447.1 ± 30.7 | 5.9 ± 0.6 | 4.1 ± 1.7 | 0.9 ± 0.4 |
| 2-V | 648.7 ± 81.5 | 9.7 ± 1.6 | 7.3 ± 3.0 | 1.1 ± 0.5 |
| 3-I | 948.1 ± 92.5 | 18.1 ± 2.7 | 6.4 ± 3.0 | 0.7 ± 0.3 |
| 3-II | 1178.0 ± 58.0 | 27.0 ± 2.5 | 10.4 ± 4.4 | 0.9 ± 0.4 |
| 3-III | 1276.0 ± 97.7 | 28.9 ± 3.8 | 10.7 ± 2.8 | 0.9 ± 0.2 |
| 3-IV | 1387 ± 111.9 | 34.0 ± 3.3 | 5.36 ± 1.8 | 0.4 ± 0.1 |
| 3-V | 1580.0 ± 150.7 | 39.4 ± 3.5 | 7.9 ± 5.3 | 0.5 ± 0.3 |
| 3-VI | 1897.0 ± 179.9 | 49.4 ± 7.2 | 11.1 ± 2.7 | 0.6 ± 0.2 |

\* Number of 3D digital ovules scored: 10 (stages 2-II- 3-II, 3-IV, 3-VI), 11 (stages 2-I, 3-III, 3-V), 13 (stage 2-II), 14 (stage 1-I), 28 (stage 1-II).
Values represent mean ± SD.

**Table 2.** Cell numbers and total volumes of the major ovule tissues.

| Stage[*] | Tissue | | | | | | | | | |
|---|---|---|---|---|---|---|---|---|---|---|
| | Nucellus | | Central region | | Inner integument | | Outer integument | | Funiculus | |
| | N cells | Volume (x10$^4$ µm$^3$) | N cells | Volume (x10$^4$ µm$^3$) | N cells | Volume (x10$^4$ µm$^3$) | N cells | Volume (x10$^4$ µm$^3$) | N cells | Volume (x10$^4$ µm$^3$) |
| 2-III | 58.1 ± 8.1 | 0.6 ± 0.1 | 35.0 ± 5.7 | 0.4 ± 0.08 | 23.5 ± 5.5 | 0.30 ± 0.07 | 53.8 ± 12.1 | 0.8 ± 0.2 | 153.7 ± 28.5 | 2.0 ± 0.5 |
| 2-IV | 64.6 ± 5.9 | 0.7 ± 0.1 | 57.6 ± 9.7 | 0.7 ± 0.01 | 48.1 ± 5.9 | 0.62 ± 0.08 | 66.6 ± 10.6 | 1.4 ± 0.3 | 210.2 ± 23.6 | 2.5 ± 0.3 |
| 2-V | 64.5 ± 10.5 | 0.7 ± 0.1 | 85.0 ± 12.2 | 1.2 ± 0.2 | 79.3 ± 8.4 | 1.0 ± 0.1 | 124.0 ± 20.2 | 3.4 ± 0.6 | 295.9 ± 50.0 | 3.4 ± 0.8 |
| 3-I | 66.9 ± 13.4 | 0.7 ± 0.1 | 118.9 ± 25.4 | 1.9 ± 0.4 | 132.6 ± 18.2 | 2.04 ± 0.4 | 235.9 ± 37.6 | 8.8 ± 1.7 | 392.4 ± 36.4 | 4.5 ± 0.6 |
| 3-II | 56.0 ± 14.4 | 0.7 ± 0.1 | 158.1 ± 16.6 | 2.9 ± 0.3 | 194.7 ± 22.9 | 4.0 ± 0.6 | 289.1 ± 12.6 | 13.2 ± 1.3 | 477.1 ± 54.2 | 6.0 ± 0.8 |
| 3-III | 64.4 ± 16.6 | 0.8 ± 0.1 | 172.2 ± 23.4 | 3.1 ± 0.5 | 216.8 ± 25.4 | 4.7 ± 0.8 | 324.5 ± 21.4 | 13.9 ± 1.8 | 496.2 ± 50.6 | 6.0 ± 1.0 |
| 3-IV | 77.7 ± 12.2 | 1.0 ± 0.2 | 191.3 ± 20.1 | 3.6 ± 0.5 | 255.8 ± 28.4 | 5.8 ± 0.6 | 349.6 ± 26.6 | 15.9 ± 1.3 | 511.1 ± 75.8 | 7.3 ± 1.7 |
| 3-V | 77.4 ± 20.4 | 1.2 ± 0.3 | 209.2 ± 31.7 | 3.9 ± 0.7 | 343.5 ± 48.3 | 7.6 ± 1.1 | 439.8 ± 57.0 | 18.6 ± 1.3 | 506.8 ± 40.1 | 7.1 ± 0.8 |
| 3-VI | 85.9 ± 17.2 | 1.3 ± 0.2 | 268.5 ± 31.4 | 5.3 ± 0.8 | 453.9 ± 69.8 | 10.6 ± 1.9 | 551.6 ± 62.7 | 22.7 ± 3.7 | 533.0 ± 68.6 | 7.7 ± 1.1 |

[*]Number of 3D digital ovules scored: 10 (stages 2-II- 3-II, 3-IV, 3-VI), 11 (stages 3-III, 3-V).

Values represent mean ± SD.

or more cells (*Figure 5A,C*). Of the 38 digital ovules spanning stage 1 only two cases with fewer than 50 cells showed either one (1048_A, 32 cells) or two (805_E, 48 cells) cells expressing pWUS signal, while all ovules with 50 or more cells showed pWUS signal. The morphological distinction

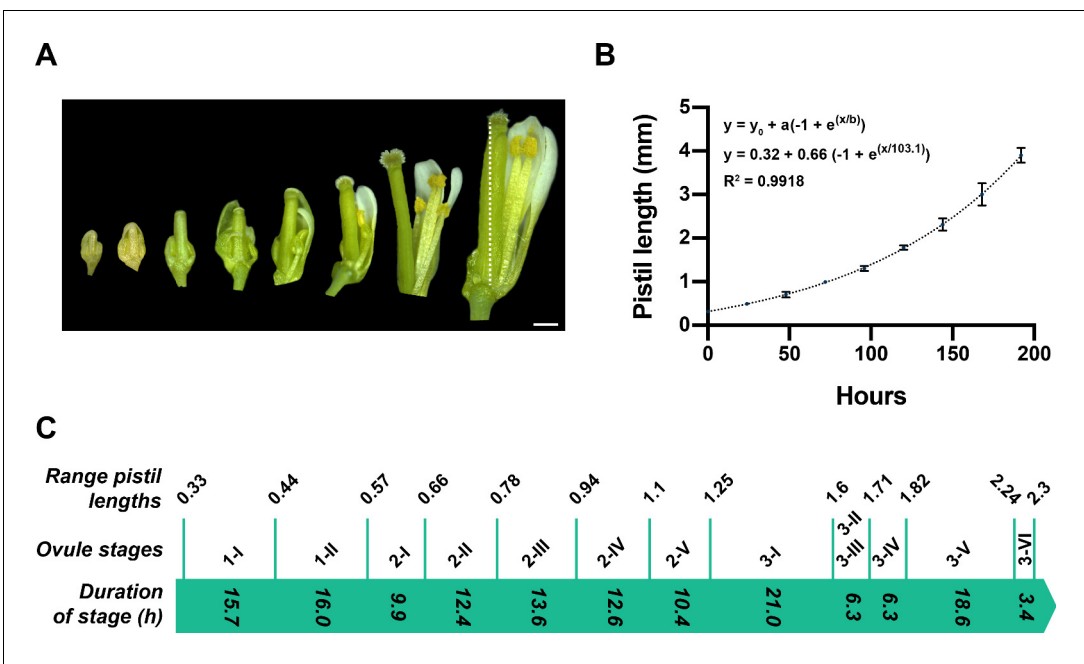

**Figure 3.** Temporal progression of ovule development. (**A**) Micrograph depicting gynoecia (pistils) at different time points (day 0 to day 7). Number of pistils scored: 6. (**B**) Graph showing the increase in pistil length (mm) over time. Mean ± SD are represented as bars. The fitted curve is indicated by the dotted line. The model and best-fit parameters are indicated. (**C**) Correlation between pistil length and duration of ovule stages. The values at the top represent the range of pistil lengths spanning the given ovule stage indicated beneath. The respective time values were deduced from the fitted curve in (**B**). The values within the green arrow indicate the deduced durations of the respective ovule stages. Number of pistils scored: 64. Scale bar: 0.5 mm.

The online version of this article includes the following source data for figure 3:

**Source data 1.** Includes the information on pistil lengths measured for each stage and the calculated duration in hours of ovule stages.

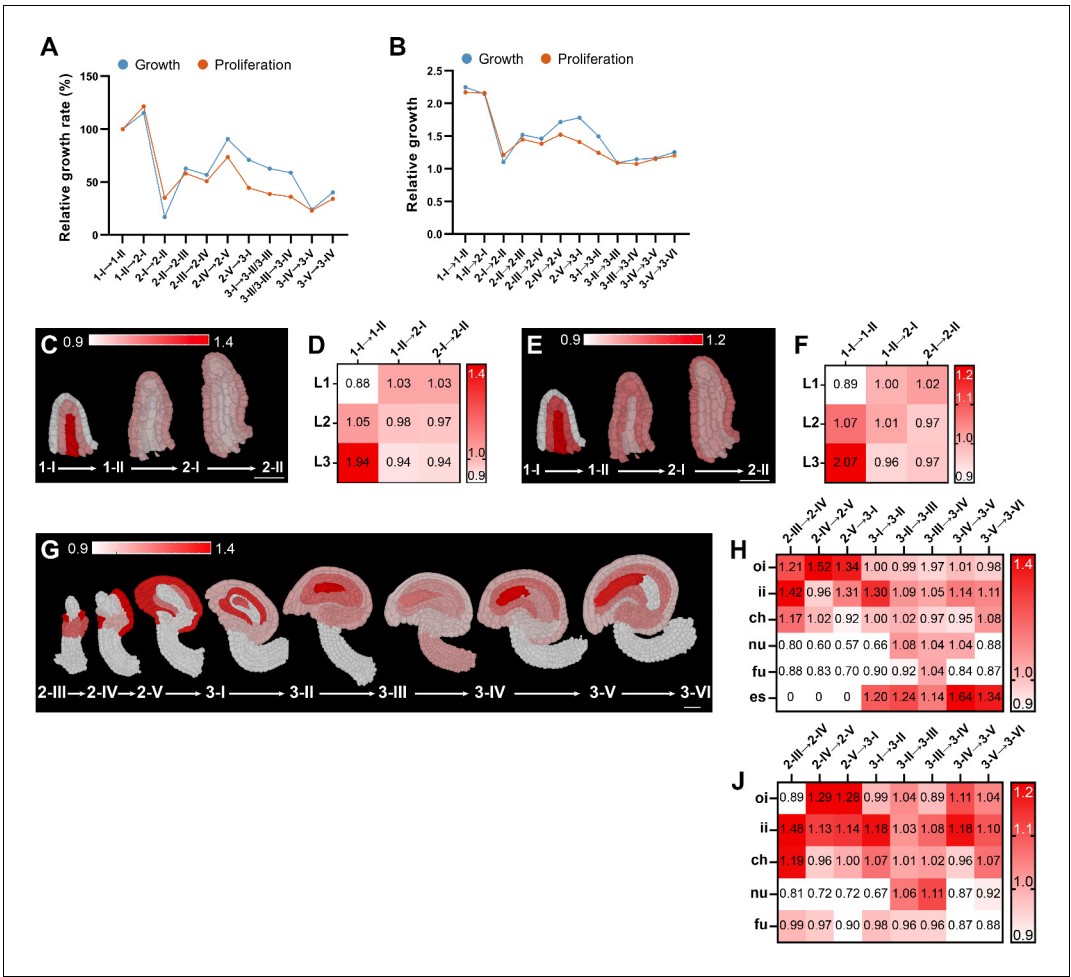

**Figure 4.** Growth dynamics of Arabidopsis ovule development. (**A, B**) Plots indicate the global volume or cell number increases. (**A**) Plot shows growth rate normalized to the growth rate of the transition from stage 1-I to 1-II. (**B**) Plot depicts relative growth between two consecutive stages. (**C–J**) Optical mid-sections and heat maps depicting relative tissue growth across the different ovule stages. Stages are indicated. Heatmap values indicate ratios. (**C, D, G, H**) Tissue-specific growth rate. (**E, F, I, J**) Tissue-specific cell proliferation rate. (**A,B**) Number of 3D digital ovules scored: 10 (stages 2-III, 2-IV, 2-V, 3-I, 3-II, 3-IV, 3-VI), 11 (3-III, 3-V), 13 (stage 2-II), 23 (stage 1-I), 49 (stage 2-I), 66 (stage 1-II). (**C,F**) 10 (stages 2-III, 2-IV, 2-V, 3-I, 3-II, 3-IV, 3-VI), 11 (stages 2-I, 3-III, 3-V), 13 (stage 2-II), 14 (stage 1-I), 28 (stage 1-II). (**G,J**) Number of 3D digital ovules scored: 10 (stages 2-III, 2-IV, 2-V, 3-I, 3-II, 3-IV, 3-VI), 11 (3-III, 3-V). Abbreviations: ch, chalaza; es, embryo sac; fu, funiculus; ii, inner integument; nu, nucellus; oi, outer integument.

The online version of this article includes the following source data for figure 4:

**Source data 1.** Includes the relative growth and proliferation, the relative growth and proliferation rate and the tissues relative growth per each ovule developmental stage.

between stages 1-I and 1-II had previously not been clearly defined (*Schneitz et al., 1995*). Thus, we used pWUS expression as a convenient marker to more precisely discriminate between the end of stage 1-I (ovules with fewer than 50 cells) and the start of stage 1-II (ovules with 50 or more cells).

Next, we explored the elaboration of the spatial pWUS expression pattern during early primordium development (*Figure 5C–F*). Reporter signal was initially detected in the epidermis of the distal tip of the primordium in individual cells or in small irregular patches of cells exhibiting reporter expression. The patchiness of the signal continued through stage 2-II. By stage 2-III, however, most of the epidermal cells of the nucellus exhibited pWUS signal. A pWUS reporter signal in the nucellar epidermis at early stage 2 is consistent with previous reports (*Tucker et al., 2012*; *Zhao et al., 2017*). However, we noticed that starting with stage 2-I about half of the ovules also exhibited

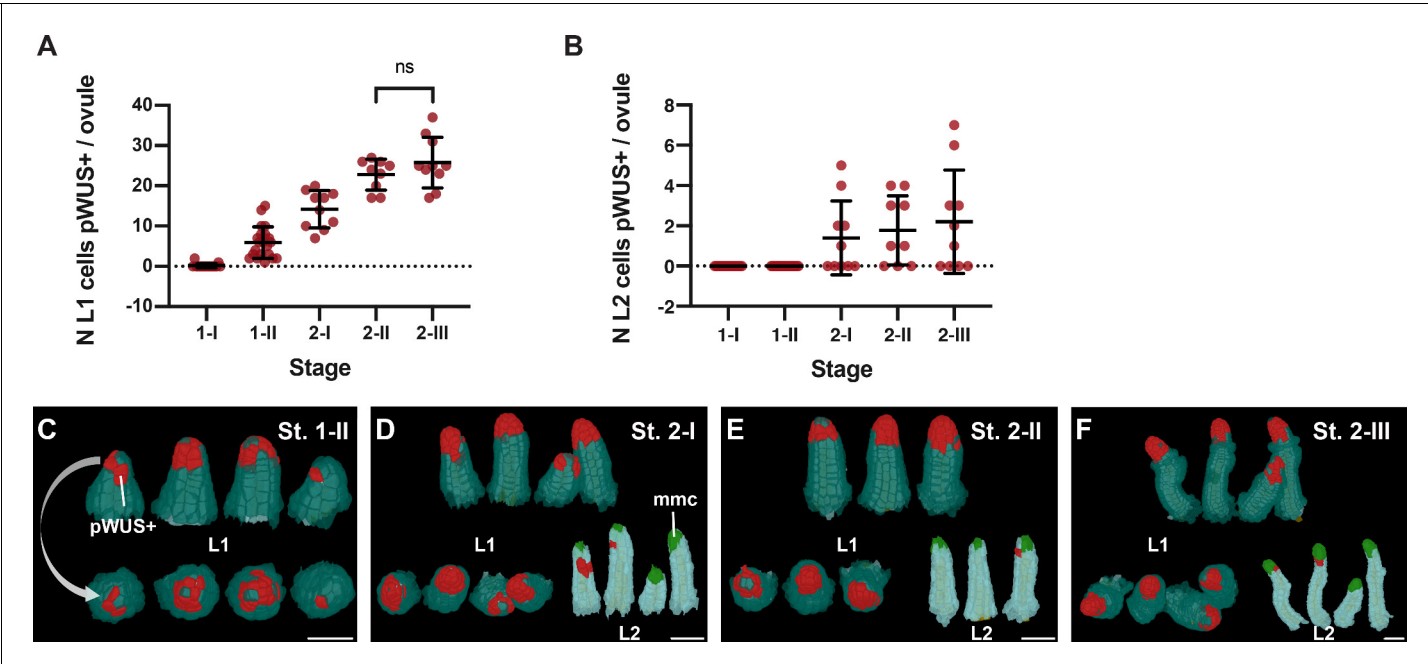

**Figure 5.** Expression pattern of the pWUS reporter. (**A**) Plot showing the number of L1 cells per ovule expressing pWUS across stage 1-I to 2-III. (**B**) Plot showing the number of L2 cells per ovule expressing pWUS across stages 1-I to 2-III. (**C–F**) 3D cell meshes displaying L1 and L2 cells expressing pWUS in red from stage 1-II to stage 2-III. Data points indicate individual ovules. Number of 3D pWUS digital ovules scored: 17 (stage 1-I), 21 (stage 1-II), 10 (stage 2-I), 9 (stage 2-II), 10 (stage 2-III). Mean ± SD are represented as bars. Scale bars: 20 μm.

The online version of this article includes the following source data for figure 5:

**Source data 1.** Includes the list of ovule IDs, stage, total number of cells, number of cells per L1, L2, L3 layer, total number of WUS expressing cells and number of WUS expressing cells per L1, L2, L3 layer of the available pWUS dataset.

reporter signal in a few cells of the first subepidermal layer (L2) (*Figure 5B,D–F*). In those instances between one to seven L2 cells were found to express the reporter. All the pWUS-expressing L2 cells were neighboring the MMC and resided next to the bottom of the MMC. We never observed reporter signal in the MMC itself. Finally, we never observed reporter signals in the central region or the integuments.

In summary, our expression study of the pWUS reporter line provided new insight into the temporal and spatial establishment of *WUS* expression in the ovule. The data suggest strong temporal control of the initiation of *WUS* expression at early stage 1-II indicating that *WUS* is not involved in ovule formation prior to the 50-cell primordium. They further suggest that *WUS* expression is mostly, though not entirely, restricted to the nucellar epidermis. Spatial regulation of *WUS* within the nucellus appears less strict as indicated by the variable and spotty expression of pWUS in the nucellar epidermis during stages 1-II to 2-I and the occasional signal in L2 cells of the nucellus from stage 2-I onwards. Our data also provide additional support for the current notion that *WUS* affects MMC formation, pattern formation in the chalaza, and integument initiation in a non-cell-autonomous fashion.

## Even growth of ovule primordia

To gain more detailed insight into specific aspects of ovule development, we explored a range of stage and tissue-specific cellular properties of the 3D digital ovules. For example, it is not known if primordium outgrowth occurs evenly or in distinct growth phases. The results in *Figure 4A* indicate that the growth rate increases from the youngest detectable primordia to stage 2-I primordia but they do not reveal whether primordium outgrowth undergoes temporal fluctuations. To address this question, we investigated early primordium development up to stage 2-I. We generated a dataset comprising 138 digital primordia of stage 2-I or younger. This dataset included ovules from the high-quality dataset described above but also encompassed ovule primordia that did not make the cut

for the high-quality dataset as they possessed slightly higher percentages of undersegmented cells (more than 4% but less than 9% of missegmented cells per primordium). This approach increased sample size and still allowed error-free determination of cell numbers (by including information from the TO-PRO-3 channel). Moreover, we could determine the total volume of the primordium, by summing up the volumes of its constituting cells, as undersegmenation errors minimally influence this parameter. This dataset was also used to determine the PD extension of the primordia.

We ordered the individual digital primordia according to increasing PD extension (primordium height), primordium volume, or cell number (*Figure 6A–D*). We did not detect distinct subclasses of primordia but observed steady and continuous rises in primordium volume and cell number. Taken together, the data support the notion that from the youngest detectable primordia to late stage 2-I primordia outgrowth does not undergo major fluctuations.

Stage 2-I is defined by the emergence of the large L2-derived MMC at the distal tip of the primordium (*Schneitz et al., 1995*). We addressed the question whether ovule primordia reach a certain size before they develop the MMC or undergo a gradual transition between stage 1-II and stage 2-I. To this end, we determined which stage 1 ovules exhibited maximum values for the three parameters mentioned above and which stage 2-I ovules featured the smallest values (*Figure 6B–D*). Taking these considerations into account, ovule primordia grow to a volume of about $1.8 \times 10^4$ $\mu m^3$, or a cell number range of approximately 125–135 total cells, and to a height range of about 41– 43 $\mu m$ when they enter stage 2-I. For each of the three parameters, we noticed a small number of

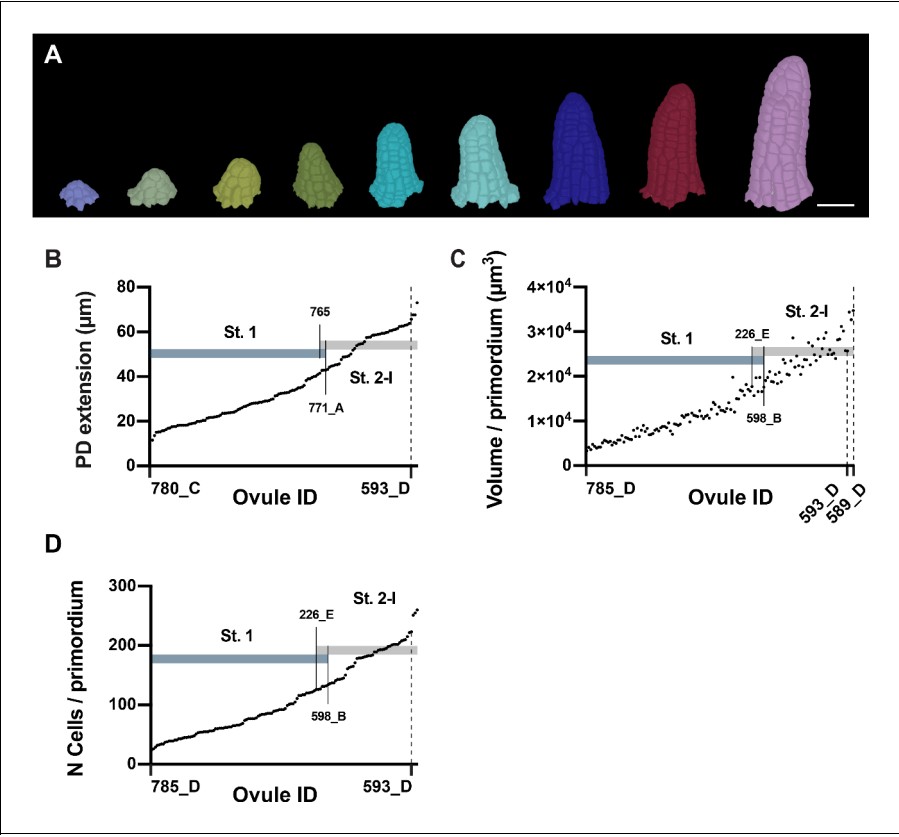

**Figure 6.** Ovule primordia grow in a continuous fashion. (A) Developmental series of 3D cell meshes of ovule primordia. (B) Plot showing an ordered array of the PD extension of ovule primordia from early initiation to the end of stage 2-I. (C) Plot indicating the total volume of primordia ordered according to increasing volume. (D) Plot depicting the total number of cells in the ovules ordered according to the increasing number of cells. Number of 3D digital ovules scored: 23 (stage 1-I), 49 (stage 2-I), 66 (stage 1-II). Scale bar: 20 $\mu m$.

The online version of this article includes the following source data for figure 6:

**Source data 1.** Includes the list of ovule IDs, total number of cells, total volume and the PD extension of early stage wild-type ovule.

ovules that fell into the range of overlap: five for the PD extension and seven each for total volume of primordium and total cell number per primordium. These numbers account for 3.6% and 5.1% of the total of 138 scored ovules, respectively. The results indicate that ovule primordia reach a size threshold before they transition from stage 1-II to stage 2-I.

Assessment of the cell volume of the L2 cells in our datasets of stage 1-II and stage 2-I digital ovules combined with visual inspection of the digital ovule primordia revealed that the MMC can easily be distinguished based on its presence in the L2 and its comparably large cell volume (*Figure 5D–F*). At stage 2-I, we measured an average MMC volume of 543 $\mu m^3$ (543.3 ± 120.6). The smallest MMC at stage 2-I possessed a volume of 339.2 $\mu m^3$, well within the range observed previously (*Lora et al., 2017*). The largest L2 cell at stage 1-II had a volume of 297.6 $\mu m^3$. We observed a mean cell volume of stage 1-II L2 cells of 153.5 $\mu m^3$ ± 48.3 (n = 335) and the volume of this large cell was well beyond the 75% percentile (187.7 $\mu m^3$). However, visual inspection revealed that this cell did not reside at the tip of the primordium while all the L2 cells located at the tip of the different stage 1-II primordia showed a cell volume that was close to the mean value. While we cannot exclude that MMC development starts already earlier than stage 2-I, we propose that by definition a stage 2-I MMC has a minimal cell volume of 335 $\mu m^3$.

## First morphological manifestation of polarity in the young ovule

The final gynapical-gynbasal orientation of the ovule relates to its alignment with respect to the apical-basal axis of the gynoecium (*Simon et al., 2012*). Here, we addressed the question when and how this orientation is set up. Upon inspecting the 3D meshes of gynoecium fragments, we noticed that many ovule primordia were positioned at a hitherto undescribed slant relative to the placenta surface (*Figure 7A*). We quantified the slant by measuring the PD distances of the shorter and longer sides of primordia of different stages (*Figure 7—figure supplement 1*). We observed that the slant was barely noticeable at stage 1-I, became more tangible during stage 1-II, and was prominent by stage 2-I (*Figure 7B,C*). Moreover, we found that the slant was oriented at a right angle to the apical-basal axis of the gynoecium with the small angle of the slant facing the developing septum. Since this axis showed a different spatial arrangement to the future gynapical-gynbasal orientation (see below), we defined the small angle half of the primordium as the posterior and the opposite half as the anterior sides of the primordium. Slant formation also coincides with anterior expression of *PHABULOSA* (*PHB*) during stage 1 (*Sieber et al., 2004*). Thus, slanting represents the first morphological manifestation of polarity in the young ovule primordium. We then investigated when and how the gynapical-gynbasal orientation of the ovule is established by looking at 40 stage 2-III and 2-IV 3D digital ovules still attached to the placenta (*Figure 7D*). We always observed initiation of the outer integument to occur at the posterior region. In ovules exhibiting slightly more advanced outer integuments we also detected a turn of the funicular PD axis (*Figure 7D*). Ovules with a noticeable turn were oriented along the apical-basal axis of the gynoecium with the anterior side pointing apically and the posterior side facing basally (*Figure 7D,E*). Thus, we propose that the final gynapical-gynbasal orientation of the ovule is the result of a multi-step morphogenetic process involving the early establishment of an anterior-posterior axis oriented normally to the long axis of the gynoecium followed by outer integument initiation and a funicular turn.

## Anterior-posterior polarity within the chalaza

Slanting and the posterior initiation of the outer integument provide anatomically recognizable signs of anterior-posterior polarity in the ovule. Slanting also predicts the curvature of the prospective funiculus (*Figure 1A*). Moreover, the shifted position of the phloem within the funiculus (*Müller et al., 2015*; *Figure 7—figure supplement 2*) and the arrangement of the nuclei in the four-nuclear embryo sac (*Figure 7—figure supplement 3A–B*) indicate a similar polarity in proximal and distal interior tissues. Thus, we investigated whether an anterior-posterior axis is also discernible within the interior central region. Previous genetic results as well as evolutionary considerations implied that the central region or chalaza can be subdivided into distal and proximal tiers flanked by the inner and outer integuments, respectively (*Baker et al., 1997*; *Endress, 2011*; *Gasser and Skinner, 2019*; *Sieber et al., 2004*). We focussed on the subepidermal cells of the proximal chalaza that constitute the majority of the central region.

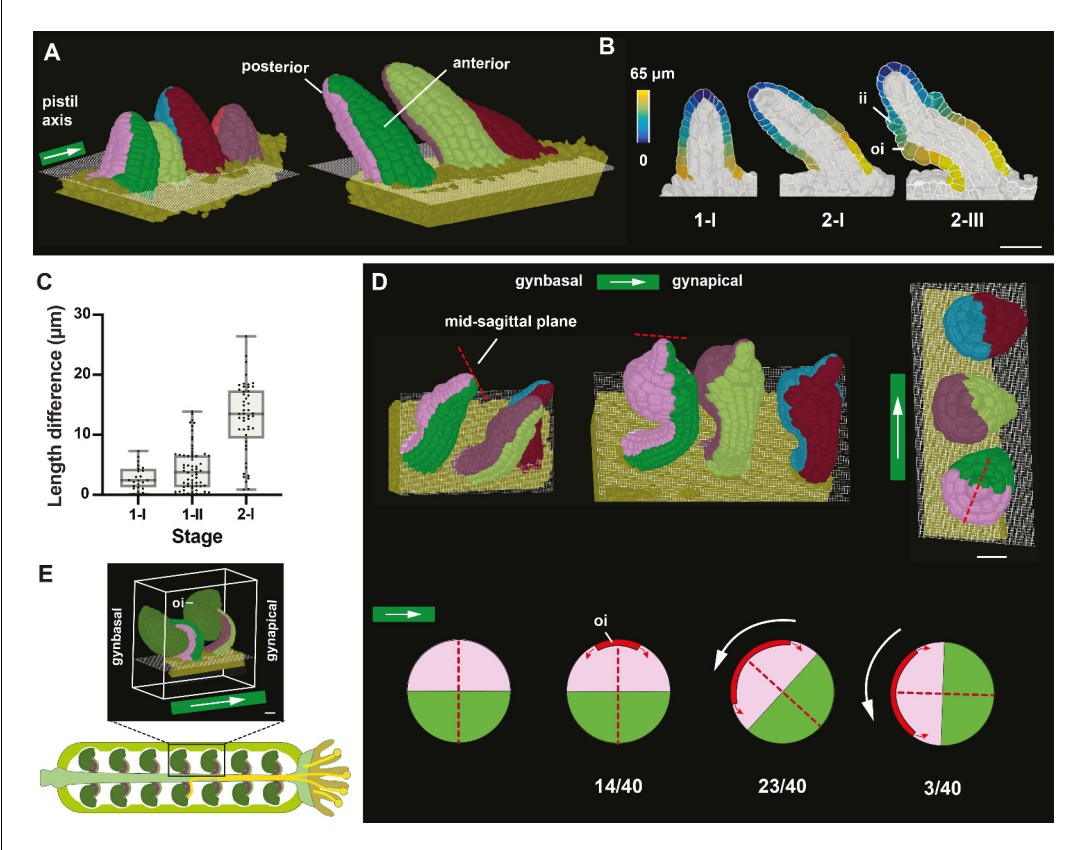

**Figure 7.** Ovule primordia slanting and early ovule polarity. (**A**) 3D meshes with multiple ovules from the same carpel attached to the placenta showing unslanted ovules at stage 1-I and slanted ovules at stage 2-I. The 2D grid represents the surface of the placenta. Color labels depict the posterior and anterior cells, respectively. The pistil axis is indicated. (**B**) 2D section views of 3D cell meshes from early ovules. Stages are indicated. The heatmap on the surface cells of posterior and anterior halves depicts the quantified distance value between individual measured cells to the distal tip of primordia. (**C**) Plot depicting the extent of slanting, quantified by the difference in maximal length on the dorsal and ventral sides of ovule at stages 1-I, 1-II, and 2-I. Data points indicate individual ovules. Mean ± SD are represented as bars. (**D**) Top row: three 3D meshes with multiple ovules attached to the placenta and highlighting the reorientation of the anterior-posterior axis relative to the apical-basal axis of the gynoecium after outer integument initiation. The right-most mesh depicts a top view of the central mesh. The dotted red line indicates the mid-sagittal plane. The white arrows highlight the pistil axis. Bottom row: cartoon depicting transverse sections of stage 2 ovules and summarising the orientations of the anterior and posterior halves of the ovule during the turn along the PD funicular axis. The relative numbers of ovules per degree of turn are given. The dotted red line indicates the mid-sagittal plane. The horizontal white arrow marks the pistil axis. (**E**) 3D view of mature ovules inside the gynoecium with the micropyle facing the apex (stigma) of the gynoecium. The posterior and anterior sides of the ovules are oriented gynbasally and gynapically, respectively. The white arrow indicates the pistil axis. Number of 3D digital ovules scored for slanting: 23 (stage 1-I), 49 (stage 2-I), 66 (stage 1-II). Number of 3D digital ovules scored for degree of turn: 8 (stage 2-IV), 32 (stage 2-III). Scale bars: 20 μm.

The online version of this article includes the following source data and figure supplement(s) for figure 7:

**Source data 1.** Includes the list of ovule IDs, the length along the anterior surface, length along the posterior surface and the length difference used for quantifying the slant of early stage wild-type ovule.

**Figure supplement 1.** Primordium length and slant measurement.

**Figure supplement 2.** Morphologically discernible polarity within the funiculus.

**Figure supplement 3.** Morphologically discernible polarity in the four-nuclear embryo sac.

Both integuments are initiated in the epidermis (*Jenik and Irish, 2000*; *Schneitz et al., 1995*). We noticed that beginning with stages 2-IV/2-V subepidermal cells restricted to the anterior side of the proximal chalaza contribute to the outer integument as well (*Figure 8A,B*). By contrast, we never found interior cells positioned more posteriorly to be part of the outer integument. In addition, cells of the abaxial or outer layer of the outer integument (oi2) extend more proximally than cells of the adaxial or inner layer of the outer integument (oi1) (*Figure 8A,B*). This interpretation of the morphology is supported by clonal analysis (see Figure 4R in *Jenik and Irish, 2000*). We defined the anterior

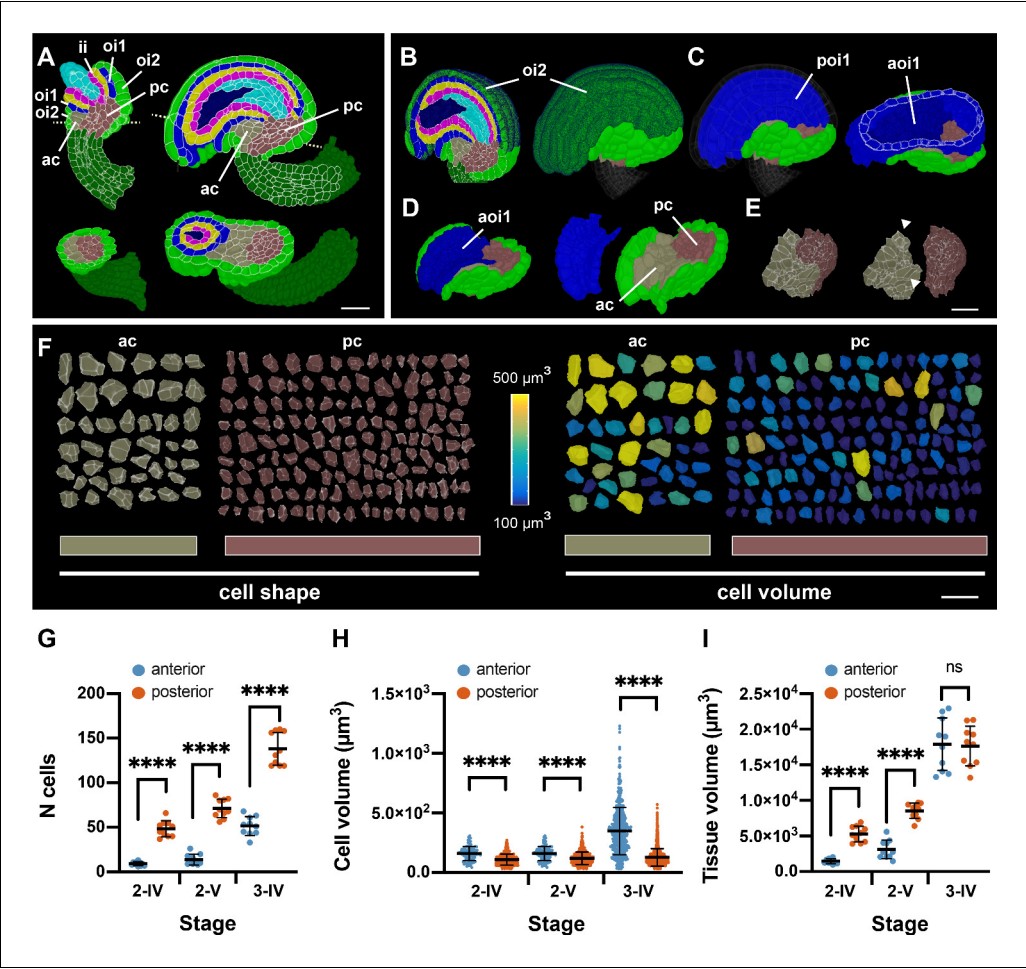

**Figure 8.** Organization of the proximal chalaza. (**A**) Left: sagittal (top) and transverse (bottom) sections through a stage 3D digital wild-type ovule. Tissues are colored as in *Figure 1*. The anterior and posterior chalaza are highlighted. The dotted lines in the upper ovule indicate the plane of the transverse section shown in the bottom ovule. Right: sagittal (top) and oblique-transverse (bottom) sections through a stage 3-IV 3D digital ovule. The dotted lines in the upper ovule indicate the plane of the oblique-transverse section shown in the bottom ovule. Note how the inner integument and the nucellus are located on top of the posterior chalaza while the anterior chalaza forms a prominent bulge. (**B–D**) Sequential removal of various tissues of the stage 3-IV ovule shown in (**A**) to reveal the anterior and posterior chalaza. (**B**) The dark-green part of the outer integument marks the extent of the oi2 layer flanking the oi1 layer and the inner integument. (**C**) The highlighted (dark-green) part of the oi2 layer in (**B**) has been removed and the posterior side of the oi1 layer is visible in blue. In the right ovule, the nucellus and inner integument were removed. A transverse section through the slightly tilted ovule now allows inspection of the anterior oi1 layer (dark blue). (**D**) The posterior oi1 layer was removed. The remaining anterior oi1 layer covers the anterior chalaza. (**E**) The anterior oi1 layer and the remaining oi2 layer were detached. The arrowheads highlight the wing-like flaps present in the anterior chalaza. (**F**) Comparisons of cell shapes and cell volumes of individual cells of the anterior and posterior chalaza, respectively. Cells are from the stage 3-IV ovule depicted in (**B**). (**G–I**) Comparison of different parameters between the anterior and posterior chalaza of stage 2-IV, 2-V, and 3-IV ovules. Data points indicate individual ovules. Mean ± SD are represented as bars. Asterisks represent statistical significance (ns, not significant; ****, p<0.0001; unpaired two-tailed t-test). (**G**) Comparison of cell numbers. (**H**) Comparison of cell volumes. (**I**) Comparison of anterior chalaza and posterior chalaza tissue volumes. Number of 3D digital ovules scored for posterior and anterior chalaza analysis: 10 (stages 2-IV, 2-V, 3-IV). Abbreviations: ac, anterior chalaza; aoi1, anterior oi1 layer; ii, inner integument; oi1, adaxial (inner) layer of outer integument; oi2, abaxial (outer) layer of outer integument; pc, posterior chalaza; poi1, posterior oi1 layer. Scale bars: 20 µm. The online version of this article includes the following video and source data for figure 8:

**Source data 1.** Includes the list of ovule IDs, stage, number of cells and tissue volume of posterior and anterior chalaza of wild-type ovule.

**Figure 8—video 1.** Anterior and posterior chalaza in a mature 3D ovule.

*Figure 8 continued on next page*

region delineated anteriorly and medio-laterally by the oi1 and oi2 layers of the outer integument as anterior chalaza. The posterior chalaza was defined as bounded medio-laterally and posteriorly by the proximal oi2 layer (*Figure 8A,B*). We found the two tissues to be composed of different numbers of cells. Moreover, the size and shape of the cells differed noticeably (*Figure 8F–I*; *Table 3*). We found that the anterior chalaza was always characterized by fewer but bigger cells in comparison to the posterior chalaza for which we observed more but smaller cells. Initially, the respective tissue volumes differed but became very similar by stage 3-I. Thus, the increases in the volumes of the anterior and posterior chalaza appear to be largely driven by increases in cell size and cell proliferation, respectively.

We noticed that despite reaching similar sizes, the two tissues exhibited differences in shape. Development of the anterior chalaza eventually resulted in a prominent anterior bulging of the outer integument likely due to the enlargement of the anterior chalazal cells. Moreover, the anterior chalaza showed two 'wing-like' lateral extensions (*Figure 8D,E*). We assume that the two 'wings' of the anterior chalaza further contribute to the bulging of the outer integument and the hood-like structure of the ovule. By contrast, the posterior chalaza acquired a more radially symmetric, rod-like shape. Starting with stages 3-I/3-II bulging of the anterior chalaza led to the appearance of the nucellus and inner integument sitting 'on top' of the posterior chalaza (*Figure 8A*).

In summary, morphological inspection of the interior chalaza of 3D digital ovules indicated that the central region is of complex composition and exhibits a recognizable anterior-posterior polarity that noticeably contributes to the shape of the ovule.

## A few scattered asymmetric cell divisions initiate a parenchymatic inner integument layer

A new cell layer is produced by the inner integument shortly before fertilization (*Debeaujon et al., 2003*; *Schneitz et al., 1995*). The ii1 or endothelial layer originates this additional cell layer (ii1') by periclinally oriented asymmetric cell divisions (*Figure 9A*). Cells of the ii1' layer are immediately distinguishable from ii1 cells by their altered shape and staining characteristics (*Schneitz et al., 1995*). Moreover, they do not express a transcriptional reporter for the epidermis-specific *ARABIDOPSIS THALIANA MERISTEM L1* (*ATML1*) gene (*Debeaujon et al., 2003*). Finally, during early seed development, ii1 cells will produce tannins while ii1' cells will contribute to parenchymatic

**Table 3.** Cellular parameters of the proximal chalaza.

| Stage[*] | Tissue | | | | | |
|---|---|---|---|---|---|---|
| | Anterior proximal chalaza | | | Posterior proximal chalaza | | |
| | N cells | Cell volume ($\mu m^3$) | Tissue volume (x$10^4$ $\mu m^3$) | N cells | Cell volume ($\mu m^3$) | Tissue volume (x$10^4$ $\mu m^3$) |
| 2-III | - | - | - | 35 ± 5 | 113 ± 41.2 | 0.4 ± 0.08 |
| 2-IV | 9 ± 2 | 160 ± 59.6 | 0.14 ± 0.03 | 48 ± 9 | 109 ± 45.6 | 0.52 ± 0.1 |
| 2-V | 13 ± 6 | 226.4 ± 77.3 | 0.31 ± 0.1 | 71.2 ± 10 | 120 ± 52.7 | 0.85 ± 0.1 |
| 3-I | 32 ± 8 | 270.4 ± 121.3 | 0.87 ± 0.2 | 86 ± 20 | 120 ± 59.5 | 1 ± 0.2 |
| 3-II | 50 ± 8 | 298 ± 156 | 1.5 ± 0.2 | 108 ± 19 | 127 ± 70 | 1.3 ± 0.2 |
| 3-III | 51 ± 13 | 318.1 ± 192 | 1.6 ± 0.35 | 121 ± 32 | 124 ± 68 | 1.5 ± 0.3 |
| 3-IV | 52 ± 11 | 347.5 ± 198.2 | 1.8 ± 0.36 | 138 ± 18 | 128 ± 73 | 1.7 ± 0.2 |
| 3-V | 60 ± 14 | 336.8 ± 227.4 | 2 ± 0.4 | 148 ± 27 | 126 ± 70.5 | 1.8 ± 0.4 |
| 3-VI | 59 ± 6 | 428.7 ± 284.4 | 2.5 ± 0.4 | 209 ± 31 | 132 ± 90.7 | 2.7 ± 0.4 |

* Number of 3D digital ovules scored: 10 (stages 2-II- 3-II, 3-IV, 3-VI), 11 (stages 3-III, 3-V).
Values represent mean ± SD.

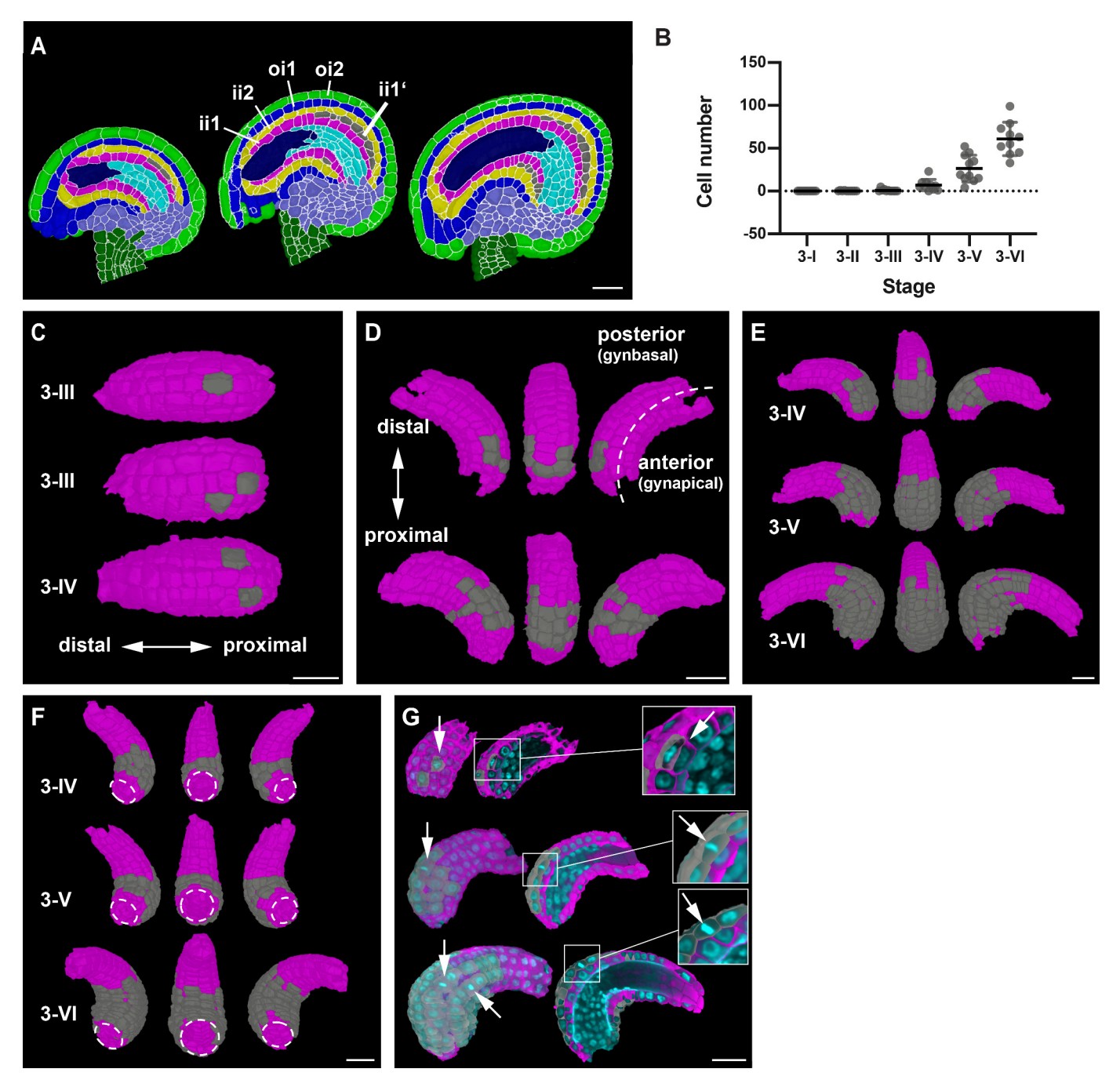

**Figure 9.** Formation of the parenchymatic inner integument layer (ii1'). (**A**) Mid-sagittal section of wild-type ovule at stages 3-IV, 3-V, and 3-VI, showing the initiation of a new cell layer (ii1') in the adaxial inner integument (endothelium/ii1). (**B**) Plot depicting the number of cells of the developing ii1' layer. Data points indicate individual ovules. Mean ± SD are represented as bars. (**C**) 3D top surface view of the adaxial inner integument at stages 3-III and 3-IV showing the occurrence of the first few pioneer cells of the ii1' layer. (**D**) 3D side surface view of adaxial inner integument depicting the pattern of occurrence of ii1' cells. (**E**) 3D side surface view of adaxial inner integument at later stages of 3-IV, 3-V, and 3-VI where the emergent tissue layer is observed to be a patch of connected cells present only at the proximal region of the inner integument. (**F**) 3D bottom surface view of ii1' layer at different stages highlighting the formation of a ring-like structure of connected cells covering the proximal half of the inner integument. (**G**) Section view of the 3D cell meshes of adaxial inner integument with the overlaid nuclei z-stack displaying a periclinal division (top) and anticlinal divisions (center, bottom) in the ii1' layer. Number of 3D digital ovules scored: 10 (stages 3-I, 3-II, 3-IV, 3-VI), 11 (stages 3-III, 3-V). Abbreviations: ii1, adaxial inner integument (endothelium); ii1', parenchymatic inner integument layer; ii2, abaxial inner integument; oi1, adaxial outer integument; oi2, abaxial outer integument. Scale bars: 20 μm.

*Figure 9 continued on next page*

*Figure 9 continued*

The online version of this article includes the following source data for figure 9:

**Source data 1.** Includes the list of ovule IDs, stage, number of cells and tissue volume of parenchymatic inner integument of wild-type ovule.

ground tissue. Thus, cambium-like activity of the ii1 layer results in asymmetric cell divisions, thickening of the inner integument, and formation of distinct tissues with separate functions.

The cellular basis and 3D architecture of ii1' layer formation is poorly understood. We followed its formation through all of stage 3 with the help of our 3D digital ovules (*Figure 9B–G*). To aid the analysis, we removed the outer integument and layer ii2 in MGX. We could observe the ii1' layer at various stages of development in 36 digital ovules. In contrast to what was previously described we already observed first signs of ii1' formation at stage 3-II (*Figure 9B,C*). Two out of the 10 digital ovules showed one cell of this layer. At stage 3-III 4/11 and at stage 3-VI 9/10, digital ovules showed at least one ii1' layer cell, respectively. By stages 3-V and 3-VI, all digital ovules exhibited this layer. The number of ii1' layer cells increased to 60.8 ± 19.5 at stage 3-VI. Thus, at this stage the average ii1' layer consisted of 13.4% of the total cells of the inner integument and with a volume of $1.53 \times 10^4$ μm$^3 \pm 0.5 \times 10^4$ μm$^3$ contributed 14.4% to its total volume.

With respect to the spatial organization of the ii1' layer, we observed single cells or patches of cells that were located on both lateral sides of the posterior inner integument at stages 3-II and 3-III (*Figure 9C*). Due to further divisions a patch of connected cells became visible at later stages. Proximal-distal and lateral extension of the ii1' layer occurred until it showed a ring-like structure covering the proximal half of the inner integument at stage 3-VI (*Figure 9D–F*). The boundary of the ii1' layer was not smooth but exhibited an irregular appearance.

We then asked if the ii1' layer is generated entirely through periclinal cell divisions of ii1 cells or if anticlinal cell divisions in existing ii1' cells contribute to the formation of this layer. To this end, we scored 978 cells of the ii1' layers across the 36 ovules exhibiting a ii1' layer and identified 14 cells in mitosis as indicated by the presence of mitotic figures in the TO-PRO-3 channel (*Figure 9G*). Only one of the 14 cells showed a periclinal cell division that generated a cell of the ii1' layer. Interestingly, however, the other 13 mitotic cells were experiencing anticlinal cell divisions.

Taken together, the results suggest that initiation of the ii1' layer does not involve simultaneous asymmetric cell divisions in many ii1 cells resulting in the formation of a ring or large patch of ii1' tissue. Rather, asymmetric periclinal cell divisions occur only in a few spatially scattered founder cells distributed within the ii1 cell layer. Further anticlinal cell divisions in the ii1' daughters of the founder cells then result in the formation of a continuous ii1' cell layer with irregular edges.

### *INNER NO OUTER* affects development of the nucellus

The YABBY transcription factor gene *INNER NO OUTER* (*INO*, AT1G23420) is an essential regulator of early pattern formation in the ovule (*Baker et al., 1997*; *Balasubramanian and Schneitz, 2002*; *Balasubramanian and Schneitz, 2000*; *Meister et al., 2002*; *Schneitz et al., 1997*; *Sieber et al., 2004*; *Villanueva et al., 1999*). The ovule phenotype of plants with a defect in *INO* has been extensively characterized. Loss-of-function *ino* mutants carry ovules that fail to form an outer integument. Moreover, development of the female gametophyte is usually blocked at the mono-nuclear embryo sac stage. However, it remains unclear if the inner integument and the nucellus, apart from the defect in embryo sac development, are affected in *ino* mutants. To address these and other issues (see below), we generated *ino-5*, a putative null allele of *INO* in Col-0 that was induced by a CRISPR/Cas9-based approach (see Materials and methods) and performed a quantitative phenotypic analysis of *ino-5* ovules using a dataset of 120 3D digital *ino-5* ovules covering stages 1-II to 3-VI ($3 \leq n \leq 20$ per stage) (*Figure 10A*). Ovules lacking *INO* activity are difficult to stage as they lack many of the distinct criteria that define the different stages of wild-type ovule development. To circumvent this problem, we staged *ino-5* ovules by comparing the total number of cells and the total volume of *ino-5* 3D digital ovules to the corresponding stage-specific values of wild-type 3D digital ovules for which the outer integument had been removed (*Supplementary file 4*).

We first analyzed the total cell number per primordium and the total volume per primordium for *ino-5* ovules of stages 1-II to 2-II. We did not find robust differences between *ino-5* and wild type for the two parameters (*Figure 10B,C*). This was to be expected as *INO* expression became first

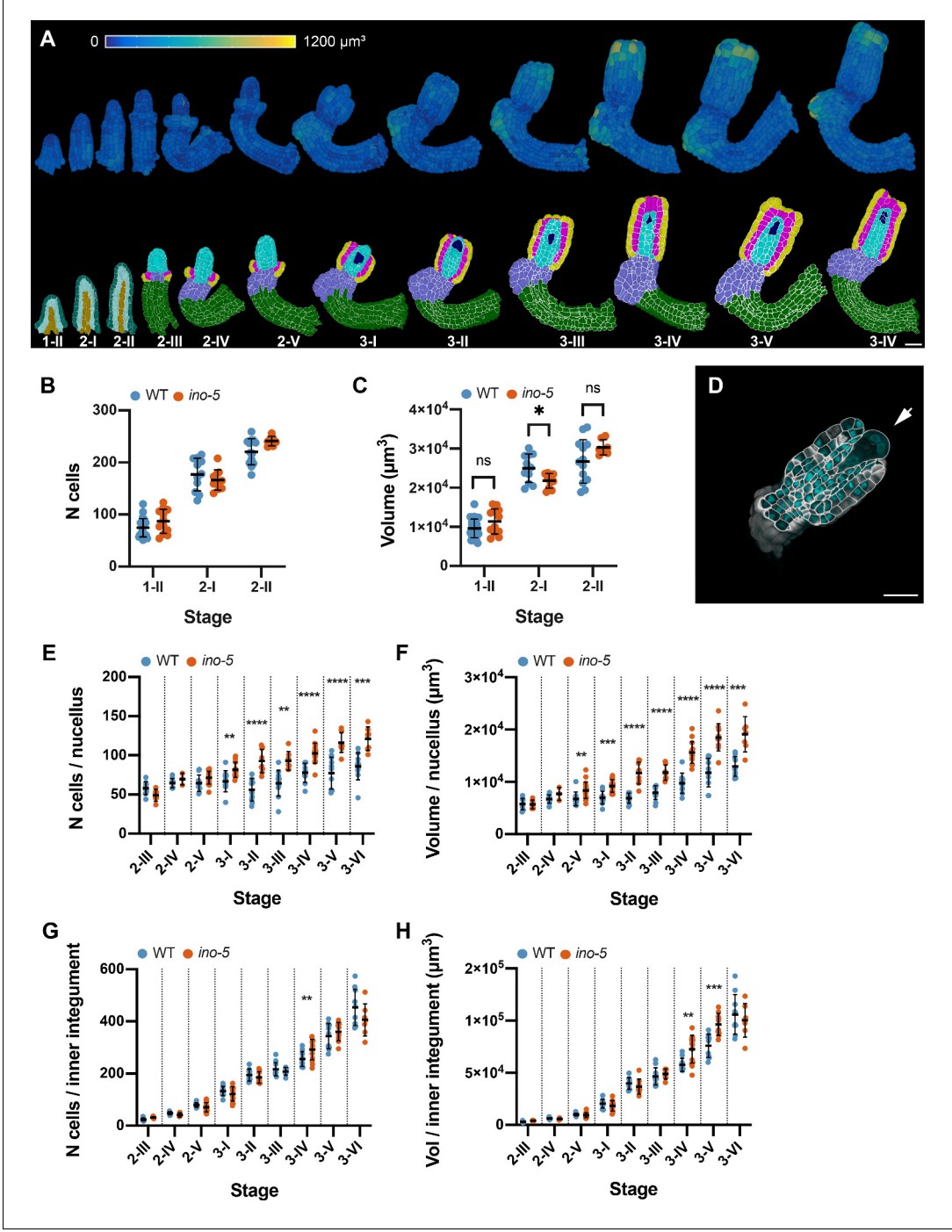

**Figure 10.** Quantitative analysis of the *ino-5* digital ovule atlas. (A) Surface view of 3D cell meshes showing heatmaps of cell volumes of *ino-5* ovules from early to late stage of development. The bottom row shows 3D mid-sagittal section views depicting the cell type organization in *ino-5* mutant ovules. (B) Plot showing the total number of cells in wild-type and *ino-5* at early stages of ovule development. (C) Plot showing the total volume of wild-type and *ino-5* ovules at early stages of ovule development. (D) Section view of the cell boundary z-stack (white) overlaid with the z-stack of stained nuclei showing a four-nuclear embryo sac in an *ino-5* ovule. (E, F) Plots comparing total cell number and tissue volume of the nucellus between wild type and *ino-5* at different stages. (G, H) Plots comparing total cell number and tissue volume of the inner integument between wild type and *ino-5* at different stages. Data points indicate individual ovules. Mean ± SD are represented as bars. Asterisks represent statistical significance (ns, p≥0.5; *, p<0.05; **, p<0.01, ***, p<0.001; ****, p<0.0001; Student's t-test). Number of

*Figure 10 continued on next page*

*Figure 10 continued*

3D digital *ino-5* ovules scored: 3 (stage 2-IV), 6 (stage 2-III), 7 (stages 2-II, 3-III, 3-VI), 9 (stages 3-II), 10 (stages 2-I, 3-V), 12 (stages 1-II), 14 (stage 3-I), 15 (stage 3-IV), 20 (stage 2-V). Scale bars: 20 μm.

The online version of this article includes the following source data for figure 10:

**Source data 1.** Includes the list of ovule IDs, stage, total number of cells and total volume of early stage *ino-5* ovules.

detectable around stage 2-II/III (*Balasubramanian and Schneitz, 2000*; *Meister et al., 2004*; *Meister et al., 2002*; *Villanueva et al., 1999*). We then asked if *INO* affects the development of the nucellus and the inner integument. To this end, we analyzed cellular features of the two tissues of *ino-5* for stages 2-III to 3-VI. We first investigated nucellar development. As previously described we observed that embryo sac development did not extend beyond the mono-nuclear embryo sac stage, although up to four-nuclear embryo sacs could be detected as well (*Figure 10D*). We then assessed cell number and tissue volume of the nucellus of *ino-5* (excluding the embryo sac). We observed an elevated number of nucellar cells in *ino-5* ovules starting from late stage 2-V/3-I (*Figure 10E*; *Tables 2* and *4*). The nucellar volume of *ino-5* was also increased relative to wild type (*Figure 10F*; *Tables 2* and *4*). The results indicate that *INO* not only affects ontogenesis of the embryo sac but also restricts cell number and nucellus volume during nucellus development.

We then investigated cell number and tissue volume for the inner integument of *ino-5* (*Figure 10G,H*; *Tables 2* and *4*). We found that the two parameters did not deviate noticeably from wild type except for stages 3-IV and 3-V where we observed a small but perceivable increase in cell number and tissue volume. However, this increase appeared to be transient as both parameters were normal in *ino-5* ovules of stage 3-VI. These findings indicate that *INO* does not exert a major influence on the monitored cellular characteristics of the inner integument.

In summary, the data suggest that *INO* exerts a hitherto unknown function in nucellus development but does not affect morphogenesis of the inner integument.

## Ovule curvature represents a multi-step process

The curvature of the Arabidopsis ovule constitutes an interesting and unique morphogenetic process. It raises the question if curvature can be subdivided into distinct, genetically controlled steps. During stages 2-III to 2-V of wild-type ovule development the central region of the ovule primordium forms a prominent kink resulting in the nucellus pointing toward the anterior side of the ovule (*Figures 1A and A*). Following kink formation differential growth results in the bending of the integuments and the nucellus resulting in the curved shape of the ovule (*Figure 1A,F*). *INO* is required for regular kink formation as this feature was absent in stage 2-V *ino-5* ovules (*Figure 11A*). In later stage *ino-5* ovules, we found aberrant growth on the anterior side of the chalaza and the

**Table 4.** Cell numbers and total volumes of different tissues in *ino-5*.

| | | | | Tissue | | | | |
|---|---|---|---|---|---|---|---|---|
| Stage* | Nucellus | | Inner integument | | Funiculus | | Central region | |
| | N cells | Volume (x10⁴ μm³) | N cells | Volume (x10⁴ μm³) | N cells | Volume (x10⁴ μm³) | N cells | Volume (x10⁴ μm³) |
| 2-III | 49 ± 7.6 | 0.5 ± 0.08 | 31.3 ± 3.2 | 0.4 ± 0.03 | 189.0 ± 29 | 2.4 ± 0.4 | 35.8 ± 37.1 | 0.3 ± 0.3 |
| 2-IV | 69.7 ± 7.5 | 0.7 ± 0.1 | 41.6 ± 8.3 | 0.5 ± 0.1 | 202.7 ± 25 | 2.2 ± 0.4 | 68.3 ± 6.6 | 0.6 ± 0.2 |
| 2-V | 71.4 ± 8.1 | 0.8 ± 0.1 | 70.7 ± 17.7 | 0.9 ± 0.2 | 286.4 ± 35.2 | 3.4 ± 0.5 | 108.1 ± 19.3 | 1.2 ± 0.3 |
| 3-I | 81.4 ± 9.5 | 0.9 ± 0.1 | 122.0 ± 27.3 | 1.8 ± 0.5 | 379.0 ± 29.7 | 4.7 ± 0.5 | 134.0 ± 23.1 | 1.8 ± 0.4 |
| 3-II | 92.6 ± 15.0 | 1.1 ± 0.2 | 185.1 ± 21.9 | 3.9 ± 0.7 | 402.9 ± 32.6 | 5.1 ± 0.7 | 176.9 ± 24.6 | 2.6 ± 0.4 |
| 3-III | 93.0 ± 12.0 | 1.1 ± 0.1 | 208.4 ± 14.6 | 4.8 ± 0.5 | 451.6 ± 32.7 | 5.9 ± 0.7 | 202.9 ± 24.1 | 3.2 ± 0.5 |
| 3-IV | 102.7 ± 12.8 | 1.5 ± 0.2 | 296.9 ± 43.5 | 7.3 ± 1.3 | 481.3 ± 45.7 | 6.7 ± 0.5 | 215.4 ± 30.7 | 3.6 ± 0.7 |
| 3-V | 116.3 ± 12.5 | 1.8 ± 0.2 | 360.8 ± 35.2 | 9.6 ± 1.0 | 522.6 ± 47.4 | 7.2 ± 1.1 | 239.6 ± 31.2 | 4.2 ± 0.7 |
| 3-VI | 121.1 ± 15.2 | 1.9 ± 0.3 | 406.1 ± 61.3 | 10.0 ± 1.6 | 599.6 ± 40.6 | 8.8 ± 0.8 | 254.4 ± 48.5 | 4.5 ± 0.9 |

* Number of 3D digital ovules scored: 3 (stage 2-IV), 6 (stage 2-III), 7 (stages 3-III, 3-VI), 9 (stage 3-II), 10 (stage 3-V), 14 (stages 3-I, 3-IV), 20 (stage 2-V ). Values represent mean ± SD.

nucellus pointing toward the posterior side of the ovule (*Figure 11A*). Apart from the absence of the regularly oriented kink in *ino* ovules, there is an obvious effect on the curvature of the ovule since the nucellus and inner integument develop into straight rather than curved structures. These observations indicated that *INO*-dependent differential growth patterns in the central region underlie kink formation.

To understand the cellular basis of early kink formation and how *INO* controls this aspect, we investigated the cellular architecture in the subepidermal central region of wild type and *ino-5*. At stage 2-III, we counted 6.7 ± 2.3 periclinal cell division planes in the subepidermal proximal chalaza (*Figure 11B,C*). This number increased to 12.6 ± 3.3 by stage 2-IV. We also observed a small number of division planes oriented in a longitudinal-anticlinal fashion (3.2. ± 2.2 at stage 2-III and 1.8. ± 2.1 at stage 2-IV) (*Figure 11B,D*). These division planes were distributed throughout the subepidermal proximal chalaza. Interestingly, we also found a few oblique division planes (1.3 ± 1.5 at stage 2-III and 1.4 ± 0.9 at stage 2-IV) (*Figure 11B,E*). They were preferentially found at the posterior chalaza just underlying the initiating outer integument. Moreover, we observed that the progeny of the posterior oblique cell divisions had undergone asymmetric enlargement with the noticeably bigger cell being located directly adjacent to the large epidermal cells associated with the outer integument initiation (*Figure 11B*). We note that by stage 2-IV, a few oblique division planes were also observed at the anterior chalaza.

In *ino-5* ovules, we noticed fewer periclinal division planes at stage 2-III/IV (*Figure 11B,C*). Furthermore, we did not observe oblique division planes (*Figure 11B,E*). Interestingly, however, the number of longitudinal-anticlinal division planes was enhanced (*Figure 11B,D*). Thus, in the absence of *INO* function, fewer periclinal cell divisions take place. Moreover, symmetric longitudinal-anticlinal cell divisions occur in place of the asymmetric oblique cell divisions in the subepidermal proximal chalaza leading to the absence of the enlarged cells abutting the posterior epidermis and a failure of kink formation.

In summary, our data indicate that differential cellular growth patterns in the proximal chalaza underly kink formation. We propose that the stage 2-III oblique cell divisions accompanied by asymmetric cell enlargement that are preferentially located in the posterior chalaza contribute to early kink formation. We further propose that the periclinal and longitudinal anticlinal divisions result in a widening of the chalaza. The control of these cellular patterns requires *INO* activity. Available evidence indicates that the spatial distribution of *INO* transcripts as well as INO protein is restricted to the epidermis and eventually the oi2 layer of the outer integument (*Balasubramanian and Schneitz, 2000*; *Meister et al., 2002*; *Villanueva et al., 1999*). This notion will require further exploration using 3D digital ovules. With this caveat in mind, however, it appears that *INO* regulates kink formation in a non-cell-autonomous fashion.

## Discussion

3D digital organs represent central tools for the study of tissue morphogenesis. In the past, it was extremely time-consuming and difficult to obtain 3D digital tissue that allowed the analysis of cellular architecture and gene and protein expression data with accurate cellular resolution including interior tissues. Here, we provide proof of concept for the power and versatility of combining the recently established segmentation pipeline PlantSeg (*Wolny et al., 2020*) with ClearSee-based staining of fixed tissue (*Kurihara et al., 2015*; *Tofanelli et al., 2019*). PlantSeg was successfully applied to diverse plant organs and even to animal tissue. Tissue clearing with ClearSee is compatible with several fluorescent proteins and stains and the optimized method employed here can be applied to a range of plant tissues as well (*Kurihara et al., 2015*; *Tofanelli et al., 2019*; *Ursache et al., 2018*). Using this approach, 3D digital ovules could be obtained with unprecedented speed and accuracy. It will be straightforward to apply this strategy to other plant tissues. Since PlantSeg provided excellent 3D cell segmentation of the *Drosophila* wing disc (*Wolny et al., 2020*) we suggest that combining PlantSeg with modern clearing methods, such as CLARITY (*Tomer et al., 2014*), may also provide a promising new strategy for the generation of 3D digital animal tissues.

This work demonstrates the unparalleled analytical power inherent in the examination of 3D digital organs with full cellular resolution. Investigating 3D digital ovules yielded new insights that could only be obtained when studying the morphogenesis of an organ with such a complex architecture in 3D and with cellular resolution throughout all tissues. For example, the analysis enabled the global

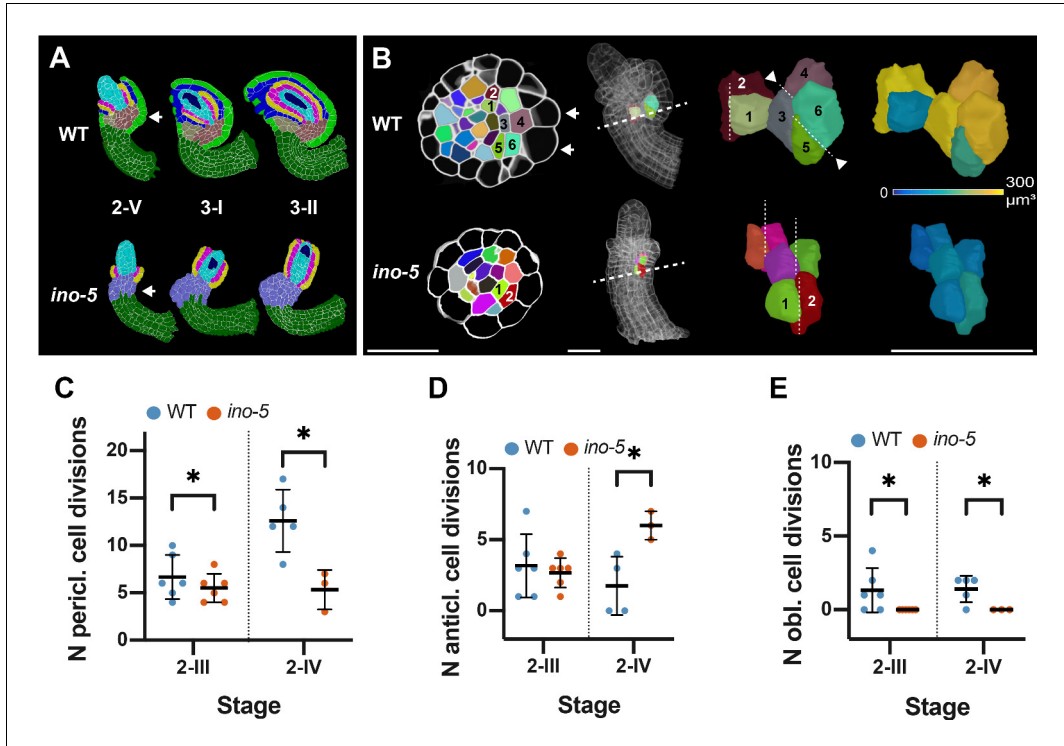

**Figure 11.** Growth patterns forming the subepidermal central region in wild-type and *ino-5* ovules. (A) Mid-sagittal section view of the cell-type-labeled 3D cell meshes of wild type and *ino-5* showing the differences in tissue organization across stages 2-V to 3-II. The arrow indicates the posterior kink in wild type and its absence in *ino-5*. (B) Top row: Left side: transverse section view depicting the division patterns observed in the chalazal region in wild-type and *ino-5* ovules. The two arrows indicate enlarged epidermal cells participating in outer integument formation. The dashed line in the ovule indicates the section plane shown on the left. Right side: Oblique 3D view of the cells numbered in the transverse section view. Dashed lines indicate the cell division plane. Arrowheads highlight oblique periclinal cell division planes. The heat map indicates the cell volumes of the pair of posterior oblique dividing cells resulting in asymmetric enlargement in wild-type and symmetric longitudinal anticlinal dividing cells in *ino-5*. (C) Plot comparing the number of periclinal divisions in wild-type and *ino-5* ovules. (D) Plot comparing the number of longitudinal-anticlinal cell divisions in wild type and *ino-5*. (E) Plot comparing the number of oblique divisions in wild type and *ino-5*. Data points indicate individual cell division. Mean ± SD are represented as bars. Asterisks represent statistical significance (ns, $p \geq 0.5$; *, $p < 0.05$; **, $p < 0.01$; ***, $p < 0.001$; ****, $p < 0.0001$; Student's t-test). Number of 3D WT digital ovules scored: 6 (stage 2-III), 5 (stage 2-IV). Number of 3D *ino-5* digital ovules scored: 3 (stage 2-IV), 6 (stage 2-III). Scale bars: 20 µm.

The online version of this article includes the following source data for figure 11:

**Source data 1.** Includes the list of ovule IDs, genotype, stage, number of different division planes scored and the volume of the pair of asymmetric enlarged oblique divided cells.

---

assessment of a diverse set of 3D cellular features of the ovule, revealed new insights into the dynamic growth patterns underlying its development, and provided new information regarding the temporal and spatial control of *WUS* expression and its role in ovule development.

The results provide a new understanding of the establishment of ovule polarity. It was previously observed that ovules are oriented relative to the long axis of the gynoecium (*Simon et al., 2012*). The gynapical (micropylar) side is pointing to the stigma and the gynbasal (chalazal) side toward the base of the gynoecium. It remained unclear, however, when this polarity became established and whether it could be recognized throughout the internal tissues of the ovule. Our results indicate that the final orientation of the ovule does not correspond to how the polarity is established. The emergence of a slant in the ovule primordium represents the earliest morphological manifestation of polarity in the ovule. Importantly, this early anterior-posterior-axis was initiated at a right angle to the long axis of the gynoecium. Our anatomical and marker gene expression data further suggest that the anterior-posterior polarity is maintained during subsequent development and throughout all

major tissues along the PD axis of the ovule. This is indicated by the polar distribution of the distal two nuclei of the four-nuclear embryo sac, the morphological differences between the anterior and posterior chalaza in the proximal central region, and the posterior placement of the phloem within the funiculus. The gynbasal-gynapical orientation of the ovule is eventually achieved by a turn in the funiculus. Thus, the alignment of the ovule with the apical-basal axis of the gynoecium does not directly correspond to the establishment of anterior-posterior polarity in the primordium but rather depends on a subsequent morphogenetic process in the funiculus. A basic molecular framework underlying primordium outgrowth and integument formation has been established (*Chaudhary et al., 2018*; *Cucinotta et al., 2014*; *Gasser and Skinner, 2019*). However, it is not known what regulates anterior-posterior polarity, slanting, and funiculus twisting. The mechanism may involve *PHB* and cues from the replum-septum boundary that is positioned in the medial plane of the gynoecium (*Reyes-Olalde and de Folter, 2019*; *Roeder and Yanofsky, 2006*).

Our data broadened the repertoire of modes of how a new cell layer can be formed in plants. Cambium-like activity of the innermost layer of the inner integument (endothelium or ii1 layer) results in the formation of the ii1' layer (*Debeaujon et al., 2003*; *Schneitz et al., 1995*). Early morphological analysis in 2D led to the hypothesis that all cells located in the roughly proximal half of the ii1 layer undergo periclinal divisions resulting in the formation of the ii1' cell layer. This model conformed to the broadly supported notion that periclinal asymmetric cell divisions underlie the formation of new cell layers in plants. For example, in the eight-cell Arabidopsis embryo, all cells undergo asymmetric cell divisions and thus contribute to the formation of the epidermis (*Jürgens and Mayer, 1994*; *Yoshida et al., 2014*). Similarly, periclinal asymmetric cell divisions in all the radially arranged cortex/endodermis initials are thought to contribute to the formation of the cortex and endodermis cell layers of the main root (*Di Laurenzio et al., 1996*; *Dolan et al., 1993*). By contrast, our investigation of 3D digital ovules revealed a distinct mode for the formation of the ii1' cell layer. The analysis indicated that only a small number of ii1 cells undergo asymmetric cell divisions and produce the first ii1' cells. Much of subsequent ii1' layer formation appears to rely on symmetric cell divisions that originate in the daughters of those few ii1' founder cells. Thus, formation of the ii1' layer largely depends on lateral extension starting from a few scattered ii1' pioneer cells, a process that may be regarded as layer invasion. It will be interesting to investigate the molecular control of this new type of cell layer formation in future studies.

3D digital organs drastically improve the power of a comparative phenotypic analysis and thus may enable new insight into gene function. As proof of concept we investigated 3D digital *ino-5* ovules. The anatomical aspects of the *ino* phenotype were thought to be reasonably well understood as several labs had carefully studied the *ino* phenotype by traditional means, such as tissue sections of paraffin-embedded specimens, optical sections through whole-mount ovules, and scanning electron microscopy (*Baker et al., 1997*; *Balasubramanian and Schneitz, 2002*; *Balasubramanian and Schneitz, 2000*; *Meister et al., 2002*; *Schneitz et al., 1997*; *Sieber et al., 2004*; *Villanueva et al., 1999*). This study, however, uncovered that *INO* plays an even more important role in the organization of the early ovule than previously appreciated. Indeed, we have found that *ino* mutants not only show a missing outer integument and defective embryo sac development but also exhibit previously unrecognized cellular aberrations in the nucellus and chalaza. It has long been recognized that ontogenesis of the embryo sac correlates with proper integument development although the molecular mechanism remains unclear (*Gasser et al., 1998*; *Grossniklaus and Schneitz, 1998*). Our new data allow an alternative interpretation involving hormonal signaling from the chalaza. Cytokinin is known to affect ovule patterning through the control of the expression of the auxin efflux carrier gene *PIN-FORMED 1* (*PIN1*) and a chalaza-localized cytokinin signal is required for early gametophyte development (*Bencivenga et al., 2012*; *Ceccato et al., 2013*; *Cheng et al., 2013*). Thus, the defects in the development of the nucellus and embryo sac of *ino-5* may relate to its cellular mis-organization of the chalaza and thus altered cytokinin and in turn auxin signaling. It will be interesting to explore this notion in future studies.

Finally, we provide new insight into the genetic and cellular processes regulating ovule curvature. Previous studies hypothesized that ovule curvature is a multi-step process with a major involvement by the proximal chalaza and the integuments (*Baker et al., 1997*; *Schneitz et al., 1997*). Here, a detailed comparison between wild-type and *ino-5* 3D digital ovules provided evidence for two distinct processes that contribute to ovule curvature: kink formation in the ovule primordium and later bending of the developing integuments. Both processes require *INO* function as neither kink

formation nor integument bending take place in *ino-5*. The data support the hypothesis that a small number of asymmetric oblique cell divisions in the stage 2-III posterior chalaza lead to the characteristic early kink of the ovule primordium. Regarding integument bending it is noteworthy that despite the straight growth of the inner integument of *ino-5* its general cellular parameters appear largely unaffected. In addition, integument bending is prominent in mutants lacking an embryo sac (*Schneitz et al., 1997*). These observations indicate that the outer integument imposes bending onto the inner integument and nucellus. Future studies will address in more detail the interplay between genetics, cellular behavior, and tissue mechanics that underlies ovule curvature.

# Materials and methods

## Key resources table

| Reagent type (species) or resource | Designation | Source or reference | Identifiers | Additional information |
|---|---|---|---|---|
| Genetic reagent (*Arabidopsis thaliana*) | *ino-5* CRISPR mutant | This paper | | In Col-0 background |
| Genetic reagent (*Arabidopsis thaliana*) | pWUS::2xVenus:NLS | *Zhao et al., 2017* | | In Col-0 background |
| Genetic reagent (*Arabidopsis thaliana*) | pPD1::GFP | *Bauby et al., 2007* | | In Col-0 background |
| Recombinant DNA reagent | sgRNA | This paper | CRISPR/Cas9 system sgRNA | ACCATCTA TTTGATC TGCCG |
| Chemical compound, drug | SR2200 | Renaissance Chemicals *Musielak et al., 2015* | | Cell wall stain |
| Chemical compound, drug | TO-PRO-3 iodide | Thermo Fisher | T3605 | Nuclear stain |
| Chemical compound, drug | Vectashield | Vectashield Laboratories | VEC-H-1000 | Antifade agent |
| Software, algorithm | MorphographX | https://morphographx.org/ | | Open source software for visualization and analysis of biological datasets |
| Software, algorithm | PlantSeg | *Wolny et al., 2020* | https://github.com/hci-unihd/plant-seg | Machine learning-based 3D image segmentation pipeline |
| Software, algorithm | GraphPad | https://www.graphpad.com/ | | Statistical analysis software |
| Software, algorithm | Adobe cc | https://www.adobe.com/ | Ai, Ps, Pr | Video and image rendering |

## Datasets

| Dataset | DOI |
|---|---|
| Wild-type (Col-0) 3D digital ovule dataset | S-BSST475 |
| WUS reporter 3D digital ovule dataset (pWUS::2xVenus:NLS) | S-BSST498 |
| *ino-5* 3D digital ovule dataset | S-BSST497 |
| Aggregates of all source files with cellular attributes of all ovules used in this study | S-BSST513 |

The entire 3D digital ovule datasets including their quantitative descriptions can be downloaded from the BioStudies data repository at EMBL-EBI (*Sarkans et al., 2018*) (https://www.ebi.ac.uk/bio-studies/). Each dataset contains raw cell boundaries, cell boundaries, predictions from PlantSeg, nuclei images, segmented cells as well as the annotated 3D cell meshes, and the associated attribute files in csv format. The 3D mesh files can be opened in MGX.

## Plant work, plant genetics, and plant transformation

*Arabidopsis thaliana* (L.) Heynh. var. Columbia (Col-0) was used as wild-type strain. Plants were grown as described earlier (*Fulton et al., 2009*). The Col-0 line carrying the *WUSCHEL* (*WUS*) promoter construct pWUS (pWUS::2xVENUS:NLS::tWUS) is equivalent to a previously described reporter line except that vYFP was exchanged for 2xVENUS (*Zhao et al., 2017*). The *ino-5* mutation (Col-0) was generated using a CRISPR/Cas9 system in which the egg cell-specific promoter pEC1.2 controls Cas9 expression (*Wang et al., 2015*). The single guide RNA (sgRNA) 5'-ACCATCTATTTGA TCTGCCG-3' binds to the region +34 to +55 of the *INO* coding sequence. The sgRNA was designed according to the guidelines outlined in *Xie et al., 2014*. The mutant carries a frameshift mutation at position 51 relative to the *INO* start AUG, which was verified by sequencing. The resulting predicted short INO protein comprises 78 amino acids. The first 17 amino acids correspond to INO, while amino acids 18–78 represent an aberrant amino acid sequence. Wild-type plants were transformed with different constructs using Agrobacterium strain GV3101/pMP90 (*Koncz and Schell, 1986*) and the floral dip method (*Clough and Bent, 1998*). Transgenic T1 plants were selected on Hygromycin (20 µg/ml) plates and transferred to soil for further inspection.

## Recombinant DNA work

For DNA work, standard molecular biology techniques were used. PCR fragments used for cloning were obtained using Q5 high-fidelity DNA polymerase (New England Biolabs, Frankfurt, Germany). All PCR-based constructs were sequenced.

## Clearing and staining of ovules

A detailed protocol was recently published (*Tofanelli et al., 2019*). Fixing and clearing of dissected ovules in ClearSee was done essentially as described (*Kurihara et al., 2015*). Tissue was fixed in 4% paraformaldehyde in PBS followed by two washes in PBS before transfer into the ClearSee solution (xylitol (10%, w/v), sodium deoxycholate (15%, w/v), urea (25%, w/v), in $H_2O$). Clearing was done at least overnight or for up to 2–3 days. Staining with SR2200 (Renaissance Chemicals, Selby, UK) was essentially performed as described in *Musielak et al., 2015*. Cleared tissue was washed in PBS and then put into a PBS solution containing 0.1% SR2200 and a 1/1000 dilution of TO-PRO-3 iodide (*Bink et al., 2001*; *Van Hooijdonk et al., 1994*) (Thermo Fisher Scientific) for 20 min. Tissue was washed in PBS for one minutes, transferred again to ClearSee for 20 min before mounting in Vecta-shield antifade agent (*Florijn et al., 1995*; Vectashield Laboratories, Burlingame, CA, USA).

## Microscopy and image acquisition

Confocal laser scanning microscopy of ovules stained with SR2200 and TO-PRO-3 iodide was per-formed on an upright Leica TCS SP8 X WLL2 HyVolution 2 (Leica Microsystems) equipped with GaAsP (HyD) detectors and a 63x glycerol objective (HC PL APO CS2 63x/1.30 GLYC, CORR CS2). Scan speed was at 400 Hz, the pinhole was set to 0.6 Airy units, line average between 2 and 4, and the digital zoom between 1 and 2. Laser power or gain was adjusted for z compensation to obtain an optimal z-stack. SR2200 fluorescence was excited with a 405 nm diode laser (50 mW) with a laser power ranging from 0.1% to 1.5% intensity and detected at 420–500 nm with the gain of the HyD detector set to 20. TO-PRO-3 iodide fluorescence excitation was done at 642 nm with the white-light laser, with a laser power ranging from 2% to 3.5% and detected at 655 to 720 nm, with the gain of the HyD detector set to 200. For high-quality z-stacks, 12-bit images were captured at a slice interval of 0.24 µm with optimized system resolution of 0.063 µm x 0.063 µm x 0.240 µm as final pixel size according to the Nyquist criterion. Some of the z stacks were captured with 2x downsampled voxel size of 0.125 µm x 0.125 µm x 0.24 µm where we took advantage of the PlantSeg-trained model to generate equal standard cell segmentation as with images with fine voxels. This was possible because the PlantSeg model 'generic_confocal_3d_unet' was trained on downsampled original

images and ground truths. The model now requires raw images whose voxels are scaled to the trained dataset so that it generates best cell boundary predictions. Overall, raw images captured with 2x downsampled voxels were helpful in that it simplified the rescaling step in PlantSeg and allowed us to generate raw images in less time without compromising segmentation quality. Images were adjusted for color and contrast using Adobe Photoshop CC (Adobe, San Jose, USA), GIMP (https://www.gimp.org/) or MorphographX (*Barbier de Reuille et al., 2015*) software (https://www.mpiz.mpg.de/MorphoGraphX). Image acquisition parameters for pWUS reporter line were the following: SR2200; 405 diode laser 0.10%, HyD 420–480 nm, detector gain 10. 2xVenus; 514 nm Argon laser 2%, HyD 525–550 nm, detector gain 100. TO-PRO-3; 642 nm White Laser 2%, HyD 660–720 nm, detector gain 100. In each case, sequential scanning was performed to avoid crosstalk between the spectra.

## Segmentation

### 3D cell segmentation

3D cells were segmented using PlantSeg (*Wolny et al., 2020*). Cell segmentation using boundary predictions in PlantSeg allowed us to process microscopic image z-stacks captured with coarse voxels, to minimize the segmentation errors, and to reduce the imaging time and hence laser exposure and phototoxicity to samples. The PlantSeg pipeline takes as input a batch of raw images in tiff format depicting the cell walls stained by SR2200 and outputs cell boundary predictions by a U-Net-based convolutional neural network (CNN) (*Ronneberger et al., 2015*) together with 3D cell segmentation given by partitioning of the boundary predictions. The PlantSeg pipeline is subdivided in four sequential steps. The first step consists of pre-processing the raw files. In particular, we found rescaling the raw images to match the U-Net training data crucial to achieve the best performances. This can be done semi-automatically (see below) by using the guided rescaling tool in the data pre-processing section of the PlantSeg gui. The output of the pre-processing step is saved as hierarchical data format (HDF5). The second step consists of predicting the cell boundaries. PlantSeg has built-in several pre-trained CNNs that can be chosen for different types of raw input data. For boundary prediction the 'generic_confocal_3D_unet' CNN was employed, which was trained on the Arabidopsis ovules dataset (https://osf.io/w38uf). The third step is where the actual segmentation is obtained. PlantSeg's segmentation is based on graph partitioning and one can choose between several partitioning strategies, such as Multicut (*Speth et al., 2011*) or GASP average (*Bailoni et al., 2019*). The last step deals with image post processing. This step is necessary to convert the images to the original voxel resolution and to convert the results from HDF5 back to tiff file format. A thorough experimental evaluation of the different parameters revealed that optimal results could be obtained by using voxel size 0.15 × 0.15 × 0.235 (µm, xyz) for the guided rescaling, 'generic_confocal_3D_unet' CNN for cell boundary prediction, and cell segmentation performed by GASP average and watershed in 3D (*Bailoni et al., 2019*) with default parameters. The PlantSeg's YAML configuration file with all the parameter settings for the pipeline is found in the Source Code File 1. These settings allowed for near-perfect 3D cell segmentation as only a small number of errors, such as over-segmented cells, had to be corrected by visual inspection of the segmented stack in MGX. In critical cases, this included cross-checking the TO-PRO-3 channel which included the stained nuclei. Thus, we now routinely image ovules using these settings. Image acquisition of mature ovules takes about 15 min for both channels (SR2200/cell contours; TO-PRO-3/nuclei), running the PlantSeg pipeline requires about 25 min on our computer hardware (1x Nvidia Quadro P5000 GPU), and manual correction of segmentation errors takes less than 5 min.

An alternative 3D segmentation (PlantSeg-MGX hybrid) was performed on some wild-type samples from stages 3-I, 3-II, 3-III, and 3-V. The z-stacks of raw cell wall images and PlantSeg cell boundary predictions were combined in MGX using the process "Stack/Multistack/Combine Stack/Max" to generate a merged image stack, which was further blurred twice with a radius of 0.3 × 0.3 × 0.3 using the process "Stack/Filter/Gaussian Blur Stack". The processed image was further segmented in MGX by auto-seeded ITK watershed with the default threshold of 1500 using "Stack/ITK/Segmentation/ITK Watershed Auto Seeded". PlantSeg-MGX hybrid segmentation was time consuming but the results were comparable to the graph-partitioning-based 3D segmentation of PlantSeg.

## Quantifying pWUS nuclei in the ovule

The nuclei labeled by pWUS signal (blobs) were counted using the Local Maxima process in MGX. The raw z-stacks were processed with the process "Stack/Filter/Brighten Darken" with a value ranging from 2 to 4 depending on image quality. Further Gaussian blur was applied twice with a low sigma of $0.2 \times 0.2 \times 0.2 \ \mu m^3$ using "Stack/Filter/Gaussian Blur Stack". The processed stack was used to generate local signal maxima of radius 1.5 µm in xyz and with a typical threshold of 8000 using process "Stack/Segmentation/Local Maxima". The blobs were visualized by creating 3D blob meshes using the process "Mesh/Creation/Mesh From Local Maxima". Blob size can be adjusted as required. We used a blob radius of size 1.2 µm that fits inside the nuclei. The blobs were also linked to their parent 3D cell meshes using the process "Mesh/Nucleus/Label Nuclei". The blob meshes were further linked to their parent 3D cell meshes using the MGX process "Mesh/Nucleus/Label Nuclei". This requires the 3D cell meshes to be loaded into the mesh 1 workspace and the blob meshes into the mesh 2 workspace of MGX. This process sets the cell identities of the 3D cell meshes as parents to the blob meshes. Blob numbers were determined in exported csv files that contained the Blob IDs associated with their parent cell IDs.

## Generation of 3D cell meshes and classification of cell types

3D cell meshes were generated in MGX using the segmented image stacks using the process "Mesh/Creation/Marching Cube 3D" with a cube size of 1. All the cell annotation was performed on the 3D cell meshes in MGX. A tissue surface mesh is required for the method of semi-automatic cell type labeling. The tissue surface mesh is generated from the segmented stack. The segmented stack was first gaussian blurred using the process "Stack/Filter/Gaussian Blur Stack" with a radius of 0.3 in xyz. The smooth stack was used to generate tissue surface mesh using "Mesh/Creation/Marching Cube Surface" with a cube size of 1 and threshold 1. The generated surface mesh was then smoothed several times using the process "Mesh/Structure/Smooth mesh" with 10 passes. For mature ovules cell type annotation, we used the MGX process "Mesh/Cell Atlas 3D/Ovule/Detect Cell Layer" (a modified 3DCellAtlas Meristem tool [*Montenegro-Johnson et al., 2019*]) with the 3D cell meshes in the mesh 1 workspace and the tissue surface mesh in the mesh 2 workspace using a cone angle parameter of 1.2. This process correctly classified about 60% of cells based on the layer they belong to. We manually corrected mis-annotations and labeled the rest of the cells by using the mesh tools in MGX. We used the "Select Connected Area" tool to select individual cells of different layers in 3D and proofread the cell type labeling with "Mesh/Cell Types/Classification/Set Cell Type". Each cell layer of the integuments was consecutively shaved off and proofreading was performed using 3D surface view. We further used the processes "Mesh/Cell Types/Classification" to save all labels, load labels and select cell types as required. The saved cell types csv file was reloaded onto the original 3D cell mesh and final proofreading was performed in the section view. For primordia, we manually annotated the cells using the tools in "Mesh/Cell Types/Classification".

## Exporting attributes from `MorphoGraphX` for further quantitative analysis

All quantitative cellular features were exported as attributes from MGX. The attributes included cell IDs (segmentation label of individual cells), cell type IDs (tissue annotation), and cell volume. The attributes from individual ovules were exported as csv files and merged to create long-format Excel-sheets listing all the scored attributes of all the cells from the analyzed ovules. These files are included in the downloadable datasets.

## Computer requirements

A minimal setup requires an Intel Core i9 CPU with at least 64 GB of RAM. To take full advantage of the PlantSeg pipeline requires at a minimum an 8 GB NVIDIA GPU/CUDA graphics card (for example a Geforce RTX 2080). For routine work, we use a computer with a 16 GB NVIDIA QUADRO P5000 card. We use Linux Mint as the operating system (http://www.linuxmint.com).

## Gynoecium length and temporal progression of ovule development

Gynoecium length in partially dissected flowers was measured from images taken by a stereo microscope (Leica Microsystems S Apo StereoZoom 1.0x - 8.0 x equipped with a Leica MC170 HD

camera). Except for the gynoecium most other floral organs were carefully removed with forceps. Gynoecia that showed any signs of degeneration or stress (anthocyanin production) during the measured growth period were omitted from the analysis. Gynoecium length was measured from the upper end of the gynophore to the top edge of the style (bottom of the stigma) using Fiji/ImageJ (*Schindelin et al., 2012*). Gynoecium length measurements were performed on six individual live gynoecia, each from a different plant and still connected to the gynophore and pedicel, using images captured every 24 hr for a maximum of 192 hr.

To establish the timeline of ovule development 64 individual pistils of different lengths and from several plants were collected, the pistil length determined, and the stages of the ovules present within a pistil were assessed. The number of ovules per stage for a given pistil was determined. Given the asynchrony of ovule development the stage with the largest number of ovules was defined as reference. Subsequently, the pistils with the shortest and longest lengths that harbored a given stage were identified. The corresponding time interval spanning the difference in pistil length was derived from the pistil growth curve. It provided an estimate of the duration of the given ovule stage.

## Growth analysis

Relative ovule growth rates were estimated as follows. First, the ratio of a given mean parameter of two consecutive stages was divided by the time interval required to progress between the two stages ($A = ln((x(n+1)/x(n))/t(n \rightarrow n+1))$). This time interval was determined by dividing by two the sum of the respective stage-specific time intervals. Second, to obtain the relative growth rate the growth rates of the different stage transitions were normalized against the growth rate obtained for the transition from stage 1-I to stage 1-II. Stage-specific relative growth was estimated by taking the ratio between a given mean parameter of two consecutive stages (x(n+1)/x(n)). Relative growth of a specific ovule tissue was estimated by taking the ratio of the respective mean parameter of two consecutive stages divided by the relative ovule growth $(y(n+1)/y(n))/(x(n+1)/x(n))$.

## Primordia length and slanting

Length along the surface of the anterior and posterior sides of primordia was quantified for individual primordia. We extracted a file of epidermal cells at the mid-section of two halves of the ovule and placed a Bezier grid of size 3 × 3 (xy) on the distal tip of the primordia surface. We then used the MGX process "Mesh/Heatmap/Measure 3D/Distance to Bezier" to quantify the length. The measured values are the distance from the Bezier grid to individual cell centroids through the file of connected cells. Primordia height was quantified by averaging the two values. Slanting was quantified by obtaining the difference of the maximal values at the anterior and posterior sides, respectively (*Figure 7—figure supplement 1*).

## Statistical analysis

Statistical analysis was performed using a combination of R (*R Development Core Team, 2019*) with RStudio (*R Studio Team, 2019*), the Anaconda distribution (Anaconda Software Distribution; https://anaconda.com) of the Python SciPy software stack (*Oliphant, 2007*), and PRISM8 software (GraphPad Software, San Diego, USA).

## Acknowledgements

We thank members of the Schneitz lab for helpful comments. We also thank Lynette Fulton, Thomas Greb, and Alexis Maizel for input and comments on the manuscript. We further thank Miltos Tsiantis for comments and support. We appreciate the gift of the pWUS line by Christian Wenzel and Jan Lohmann and thank Uli Hammes for the pPD1::GFP reporter line. We further acknowledge support by the Center for Advanced Light Microscopy (CALM) of the TUM School of Life Sciences. This work was funded by the German Research Council (DFG) through grants FOR2581 (TP3) to AK and FH, (TP8) to RS, and (TP7) to KS.

## Additional information

### Funding

| Funder | Grant reference number | Author |
| --- | --- | --- |
| Deutsche Forschungsge-meinschaft | FOR2581 (TP3) | Anna Kreshuk<br>Fred A Hamprecht |
| Deutsche Forschungsge-meinschaft | FOR2581 (TP8) | Richard S Smith |
| Deutsche Forschungsge-meinschaft | FOR2581 (TP7) | Kay Schneitz |

The funders had no role in study design, data collection and interpretation, or the decision to submit the work for publication.

### Author contributions

Athul Vijayan, Rachele Tofanelli, Conceptualization, Formal analysis, Investigation, Visualization, Methodology, Writing - review and editing; Sören Strauss, Lorenzo Cerrone, Adrian Wolny, Software, Investigation, Methodology; Joanna Strohmeier, Investigation; Anna Kreshuk, Fred A Hamprecht, Conceptualization, Software, Funding acquisition, Methodology; Richard S Smith, Conceptualization, Software, Supervision, Funding acquisition, Methodology, Writing - review and editing; Kay Schneitz, Conceptualization, Formal analysis, Supervision, Funding acquisition, Visualization, Writing - original draft, Project administration, Writing - review and editing

### Author ORCIDs

Athul Vijayan https://orcid.org/0000-0003-1837-6359
Rachele Tofanelli http://orcid.org/0000-0002-5196-1122
Anna Kreshuk http://orcid.org/0000-0003-1334-6388
Richard S Smith https://orcid.org/0000-0001-9220-0787
Kay Schneitz https://orcid.org/0000-0001-6688-0539

### Decision letter and Author response

Decision letter https://doi.org/10.7554/eLife.63262.sa1
Author response https://doi.org/10.7554/eLife.63262.sa2

## Additional files

### Supplementary files

- Source code 1. Yaml file with PlantSeg parameters used in this study.
- Supplementary file 1. Standardized cell type labels.
- Supplementary file 2. Pistil length of individual live pistils.
- Supplementary file 3. Correlation of pistil length with ovule stages.
- Supplementary file 4. Cell numbers and total volume of ino-5.
- Transparent reporting form

### Data availability

The entire 3D digital ovule datasets can be downloaded from the BioStudies data repository (https://www.ebi.ac.uk/biostudies/). Each dataset contains raw cell boundaries, cell boundaries, predictions from PlantSeg, nuclei images, segmented cells as well as the annotated 3D cell meshes, and the associated attribute files in csv format. The 3D mesh files can be opened in MorphoGraphX. Accession S-BSST475: the wild-type high-quality dataset and the additional dataset with more segmentation errors. Accession S-BSST498: pWUS::2xVenus:NLS. Accession S-BSST497: ino-5. Accession S-BSST513: Long-format Excel-files listing all the scored attributes of all the cells from the analyzed ovules.

The following datasets were generated:

| Author(s) | Year | Dataset title | Dataset URL | Database and Identifier |
|---|---|---|---|---|
| Vijayan A, Tofanelli R, Strauss S, Cerrone L, Wolny A, Strohmeier J, Kreshuk A, Hamprecht FA, Smith RS, Schneitz K | 2020 | WT Arabidopsis ovule atlas: A 3D digital cell atlas of wild-type Arabidopsis ovule development with cellular and tissue resolution | https://www.ebi.ac.uk/biostudies/studies/S-BSST475 | BioStudies, S-BSST475 |
| Vijayan A, Tofanelli R, Strauss S, Cerrone L, Wolny A, Strohmeier J, Kreshuk A, Hamprecht FA, Smith RS, Schneitz K | 2020 | 3D gene expression atlas of WUSCHEL in Arabidopsis ovules with cellular, nuclear and tissue resolution | https://www.ebi.ac.uk/biostudies/studies/S-BSST498 | BioStudies, S-BSST498 |
| Vijayan A, Tofanelli R, Strauss S, Cerrone L, Wolny A, Strohmeier J, Kreshuk A, Hamprecht FA, Smith RS, Schneitz K | 2020 | ino Arabidopsis ovule atlas: A 3D digital cell atlas of ino ovules in Arabidopsis with cellular and tissue resolution | https://www.ebi.ac.uk/biostudies/studies/S-BSST497 | BioStudies, S-BSST497 |
| Vijayan A, Tofanelli R, Strauss S, Cerrone L, Wolny A, Strohmeier J, Kreshuk A, Hamprecht FA, Smith RS, Schneitz K | 2020 | 3D qualitative attribute dataset of Arabidopsis ovule atlas | https://www.ebi.ac.uk/biostudies/studies/S-BSST513 | BioStudies, S-BSST513 |

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

## Appendix 1

### Quantitative analysis of major tissues and cellular patterns in integuments

Tissue-specific growth patterns along the PD axis following primordium formation

We investigated if there were tissue-specific growth differences. To this end, we counted cell numbers and total volumes in the nucellus, embryo sac, chalaza, funiculus and the epidermal cells of the two integuments of stage 2-III to 3-VI digital ovules (*Appendix 1—figure 1*, *Appendix 1—figure 2*; *Table 2*). We observed that cell numbers in the nucellus stayed roughly constant from stage 2-III up to 3-III (*Appendix 1—figure 1A*). At stage 3-I the average cell number per nucellus was $66.9 \pm 13.4$. The results suggest that little if any cell proliferation takes place during these stages in the nucellus. Starting with stage 3-IV, however, cell numbers increased, and we found an average cell number of $85.9 \pm 17.2$ at stage 3-VI hinting at somewhat elevated cell proliferation during these latter stages. This pattern was mirrored by a 2.2-fold increase in the average volume of the nucellus across different stages (excluding the developing embryo sac) (*Appendix 1—figure 1B*) (stage 2-III: $0.6 \times 10^4$ $\mu m^3 \pm 0.1 \ 10^4 \ \mu m^3$; stage 3-VI: $1.3 \times 10^4 \ \mu m^3 \pm 0.2 \times 10^4 \ \mu m^3$). We did not detect nucellar cells at the very micropylar (distal) end of the developing embryo sac (*Figure 1F*) from stage 3-IV onwards. This observation confirmed previous observations (*Schneitz et al., 1995*) and raised the question what happened to these distal nucellar cells. Although we cannot exclude that a small number of cells becomes crushed, we did not observe such events in this region. Thus, as the average cell number per nucellus stays constant or even increases slightly we propose that the growing embryo sac 'pushes away' some distal nucellar cells.

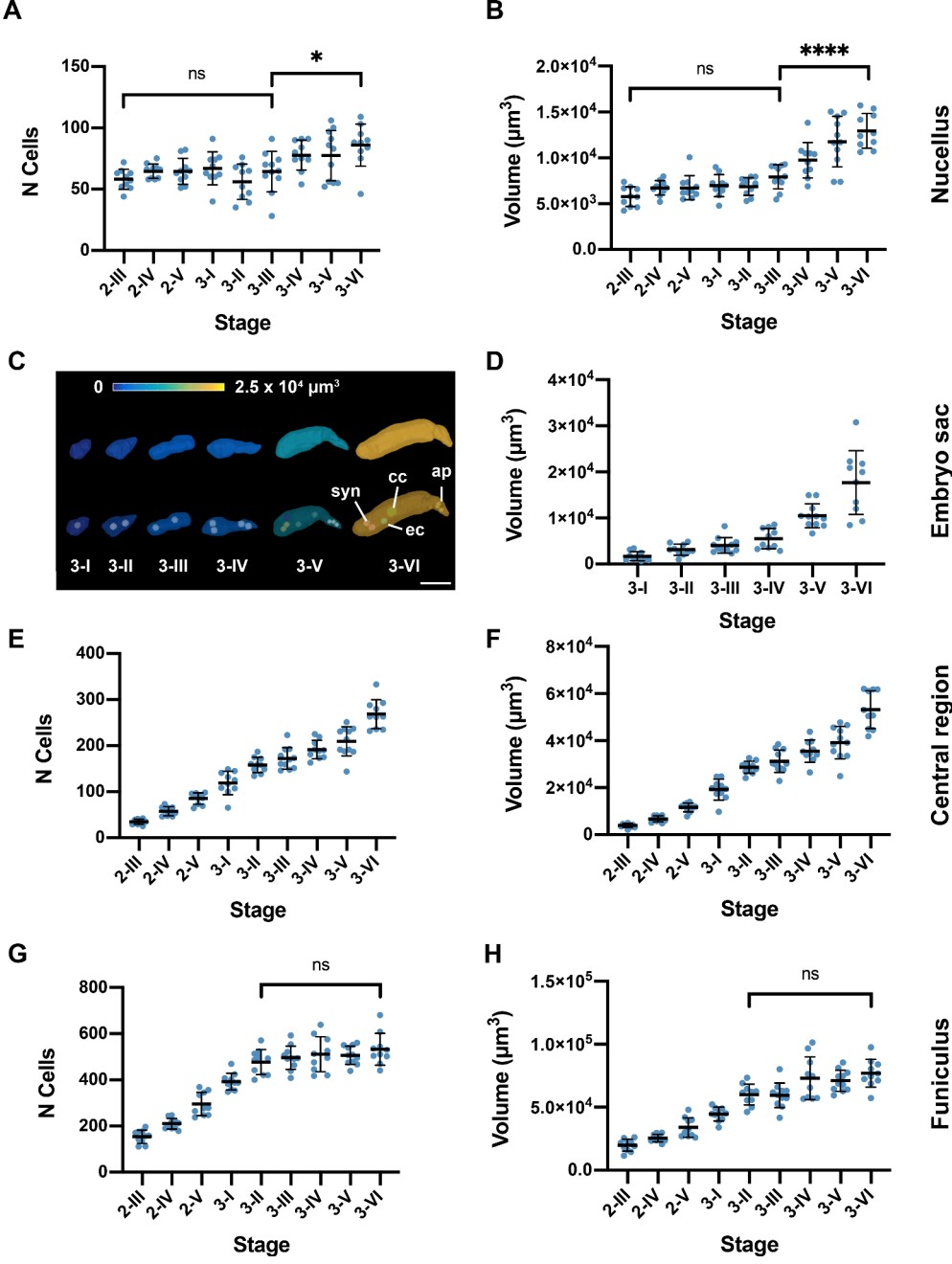

**Appendix 1—figure 1.** Tissue-specific quantitative analysis. (**A, B**) Plots depicting the number of cells and volume at different developmental stages of the nucellus, respectively. (**C**) 3D mesh of the embryo sac, from stage 3-I to stage 3-IV, extracted from 3D ovule cell meshes. The volume is represented as a heat map. (**D**) Plot depicting the embryo sac volume from individual ovule datasets at different stages. (**E, F**) Plots showing the number of cells and volume of the central region, respectively. (**G, H**) Plots depicting the number of cells and volume of the funiculus, respectively. Data points indicate individual ovules. Mean ± SD are represented as bars. Asterisks represent statistical significance (ns, p≥0.5; *, p<0.05; ****, p<0.0001; Student's t-test). Number of 3D digital ovules scored: 10 (stages 2-III, 2-IV, 2-V, 3-I, 3-II, 3-IV, 3-VI), 11 (stages 3-III, 3-V). Scale bar: 20 μm.

The online version of this article includes the following source data is available for figure 1:

**Appendix 1—figure 1—source data 1.** Includes the list of ovule IDs, stage, number of cells and tissue volume of nucellus, embryo sac, chalaza and funiculus of wild-type ovule.

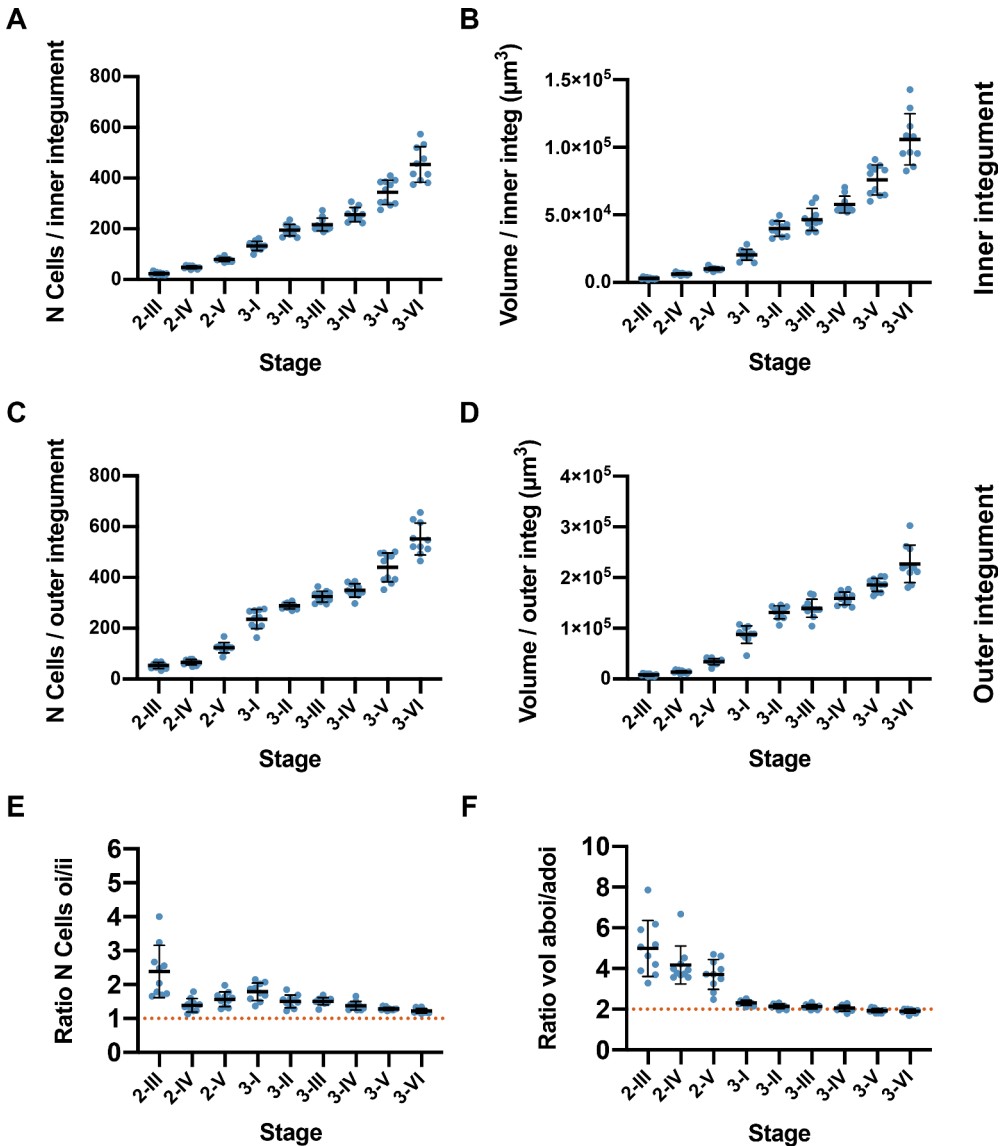

**Appendix 1—figure 2.** Quantitative analysis of cellular patterns in the integuments. (**A, B**) Plots indicating the number of cells and volume of the inner integument from stage 2-III to stage 3-VI. (**C, D**) Plots indicating the number of cells and volume of the outer integument from stage 2-III to 3-VI stages. (**E, F**) Plot showing the ratio between the number of cells and tissue volume of the outer and inner integument. Data points indicate individual ovules. Number of 3D digital ovules scored: 10 (stages 2-III, 2-IV, 2-V, 3-I, 3-II, 3-IV, 3-VI), 11 (stages 3-III, 3-V). Mean ± SD are represented as bars.

The online version of this article includes the following source data is available for figure 2:

**Appendix 1—figure 2—source data 1.** Includes the list of ovule IDs, stage, number of cells and tissue volume of outer and inner integument of wild-type ovule.

After meiosis coenocytic embryo sac development starts at stage 3-I. Three rounds of mitoses followed by cellularization eventually result in the typical eight-nuclear, seven-celled *Polygonum*-type of embryo sac found at stage 3-VI. The mono-nuclear embryo sac features a volume of $0.2 \times 10^4$ $\mu m^3 \pm 0.1 \times 10^4$ $\mu m^3$. At stage 3-VI we observed a total volume of $1.8 \times 10^4$ $\mu m^3 \pm 0.7 \times 10^4$ $\mu m^3$, a nine-fold increase (*Appendix 1—figure 1C*).

The central region of the ovule, classically known as chalaza, is of complex composition (see below). Here, we discriminate between the epidermis-derived cells of the two integuments and the internal cells of the central region encapsulated by the epidermis and first focus on the latter. We found a steady increase in the average internal cell number per central region from stage 2-III

(35.0 ± 5.7) up to stage 3-VI (268.5 ± 31.4) (*Appendix 1—figure 1E,F*; *Table 2*). We also observed a steady increase in volume of the internal central region (stage 2-III: $0.4 \times 10^4$ µm$^3$±0.1 10$^4$ µm$^3$; stage 3-VI: $5.3 \times 10^4$ µm$^3$±0.8×10$^4$ µm$^3$). Thus, cell numbers increased 7.7-fold while tissue volume increased 13.3-fold over the scored stages.

Regarding cell number in the funiculus, we observed a steady increase in the average cell number per funiculus from stage 2-III until the average cell number per funiculus reached 477.1 ± 54.2 at stage 3-II (*Appendix 1—figure 1G*; *Table 2*). This value stayed about constant throughout the later stages although we observed a minor but statistically insignificant increase to 533.0 ± 68.6 at stage 3-VI. We further observed an average volume at stage 3-VI of $7.7 \times 10^4$ µm$^3$±1.1×10$^4$ µm$^3$ (*Appendix 1—figure 1H*). The data suggest that there is very little if any growth in the funiculus after stage 3-II.

Next, we assessed the number of epidermis-derived cells of the two integuments (*Appendix 1—figure 2*; *Table 2*). Both integuments showed continuous increases in cell number and integument volume for stages 2-III to 3-VI. We observed a 19.3-fold increase in cell number from stage 2-III to 3-VI. At stage 2-III the inner integument featured an average cell number of 23.5 ± 5.5 and at stage 3-VI of 453.9 ± 69.8 (*Appendix 1—figure 2A*). Regarding the volume increase of the inner integument, we observed an 35.3-fold increase in inner integument volume with an average volume of $0.3 \times 10^4$ µm$^3$±0.07×10$^4$ µm$^3$ at stage 2-III and $10.6 \times 10^4$ µm$^3$±1.9×10$^4$ µm$^3$ at stage 3-VI (*Appendix 1—figure 2B*). For the outer integument we observed a 10.3-fold increase in cell number with an average number of cells of 53.8 ± 12.1 at stage 2-III and of 551.6 ± 62.7 at stage 3-VI (*Appendix 1—figure 2C*). Regarding the volume increase of the outer integument, we observed a 28.4-fold increase in volume with an average volume of $0.8 \times 10^4$ µm$^3$±0.02×10$^4$ µm$^3$ at stage 2-III and $22.7 \times 10^4$ µm$^3$±3.7×10$^4$ µm$^3$ at stage 3-VI (*Appendix 1—figure 2D*). We also investigated the ratio of number of cells and volume of the outer integument versus the inner integument (*Appendix 1—figure 2E, F*). The data revealed that the outer integument always carried more cells than the inner integument, although the difference was relatively small throughout the various stages with stage 3-VI showing a ratio of 1.2. By contrast, the volume of the outer integument was always more than twice the volume of the inner integument. At stage 3-I in particular we noticed a 4.3-fold larger volume of the outer integument. This value slowly decreased and at stage 3-VI we determined a ratio of 2.2.

