## [Decision Letter]

**Acceptance summary:**

This paper is a tour de force analysis of ovule growth and a showcase of how to do 3D growth analyses of complex organs to analyze all tissue layers. This temporal-spatial atlas of organ growth and development in the ovule was previously not amenable for live cell imaging due to its compound 3D structure. The resulting dataset is an important compendium of ovule growth. The authors demonstrate the power of this analysis by providing several new insights into ovule development, including a detailed analysis of cells expressing the stem cell factor *WUSCHEL*, and a detailed phenotypic characterization of the *inner no outer* mutant.

**Decision letter after peer review:**

Thank you for submitting your article "A digital 3D reference atlas reveals cellular growth patterns shaping the Arabidopsis ovule" for consideration by *eLife*. Your article has been reviewed by four peer reviewers, including Sheila McCormick as the Reviewing Editor and Reviewer #1, and the evaluation has been overseen by Christian Hardtke as the Senior Editor. The following individual involved in review of your submission has agreed to reveal their identity: Dolf Weijers (Reviewer #4).

The reviewers have discussed the reviews with one another and the Reviewing Editor has drafted this decision to help you prepare a revised submission.

Title: The title says 3D, but in the paper it says 4D.

Summary:

This paper is a tour de force analysis of ovule growth and a showcase of how to do 3D growth analyses of complex organs to analyze all tissue layers. The authors scanned serial optical sections of Arabidopsis ovules during development, from small primordia to mature organs that harbor the female gametophyte shortly before fertilization. This temporal-spatial atlas of organ growth and development in the ovule was previously not amenable for live cell imaging due to its compound 3D structure. The resulting dataset is an important compendium of ovule growth. The authors demonstrate the power of this analysis by providing several new insights into ovule development, including a detailed analysis of cells expressing the stem cell factor *WUSCHEL*, and a detailed phenotypic characterization of the inner no outer mutant. However, the authors should make more of an effort to emphasize the utility of this approach in general, we as the ovule community is small. That is, will others use similar methods to study other organs of their interest?

Furthermore, the manuscript is very long in relation to the substance presented. The paper is a tough read due to all the descriptions of cell numbers, volumes and percentages. For most of these, the significance is not clear, and we therefore strongly suggest reducing such descriptions in the main text. For example, Figures 1-5 could easily be condensed to fewer figures, and some of the graphs and quantifications could be moved to supplementary figures.

Overall, the analyses lead to interesting new insights regarding the behavior of different cell populations, the dynamics of growth, the expression of *WUS* relative to ovule development, the initiation of the 5th integument layer and the role of the INO regulator. But these are mostly snippets that could each be leads for further investigation, and are not followed to a point where we learn something entirely new. For example, the slanting growth is interesting, but what is the significance, and is there any idea as to what regulates it? The *WUS* expression pattern is interesting, but not connected at all to phenotypes of the wus mutant, hence it is unclear what lesson to take from this. The continuous growth aspect is interesting, but it is not clear why this was tested. Was the hypothesis that there are growth pulses? Is the result unexpected?

The interesting description of the ino mutant phenotype, and any discussion of the non-autonomous role in promoting inner cell growth and division, is only meaningful if INO protein accumulation is also shown in the same ovules, as was done for *WUS*.

The relevance of the identification of cell populations in the chalazal domain (Figure 12B/C) is not clear. Also, the definition of what cells belong to which domain seems fairly arbitrary. What criteria were used? There is a risk of circular argumentation here.

In sum, please make more of a clear case for how the pipeline and atlas can be used to discover new functions of known genes, or better explain a mutant phenotype.

Revisions expected in follow-up work:

For example, localizing the INO protein in segmented ovules and/or a description of the *wus* phenotype using this pipeline would add more biological insight.

---

## [Author Response]

Title: The title says 3D, but in the paper it says 4D.

This has been amended.

Summary:This paper is a tour de force analysis of ovule growth and a showcase of how to do 3D growth analyses of complex organs to analyze all tissue layers.The authors scanned serial optical sections of Arabidopsis ovules during development, from small primordia to mature organs that harbor the female gametophyte shortly before fertilization. This temporal-spatial atlas of organ growth and development in the ovule was previously not amenable for live cell imaging due to its compound 3D structure. The resulting dataset is an important compendium of ovule growth. The authors demonstrate the power of this analysis by providing several new insights into ovule development, including a detailed analysis of cells expressing the stem cell factor WUSCHEL, and a detailed phenotypic characterization of the inner no outer mutant. However, the authors should make more of an effort to emphasize the utility of this approach in general, we as the ovule community is small. That is, will others use similar methods to study other organs of their interest?

In the Discussion of the revised manuscript, we now better emphasize the general utility of the approach. The PlantSeg pipeline used for cell outline prediction and segmentation was already shown to be applicable to many different plant and even animal tissues. Moreover, the staining methods invoked in this study can also be employed for different plant organs.

Furthermore, the manuscript is very long in relation to the substance presented. The paper is a tough read due to all the descriptions of cell numbers, volumes and percentages. For most of these, the significance is not clear, and we therefore strongly suggest reducing such descriptions in the main text. For example, Figures 1-5 could easily be condensed to fewer figures, and some of the graphs and quantifications could be moved to supplementary figures.

We considerably reorganized the manuscript to accommodate the raised issues. Some data were eliminated and most of the mentioned lengthy descriptions were moved to Appendix 1. Moreover, we included additional data of more general relevance and streamlined the presentation and discussion of the results that remained in the manuscript. Overall, we believe that the revised manuscript provides more substance and is much easier to read.

Revisions for this paper:Overall, the analyses lead to interesting new insights regarding the behavior of different cell populations, the dynamics of growth, the expression of WUS relative to ovule development, the initiation of the 5th integument layer and the role of the INO regulator. But these are mostly snippets that could each be leads for further investigation, and are not followed to a point where we learn something entirely new. For example, the slanting growth is interesting, but what is the significance, and is there any idea as to what regulates it?

In this study we established that early slanting represents the first morphological manifestation of polarity in the developing ovule. This is a new finding. The respective anterior-posterior polarity is subsequently observed throughout ovule development and in all (also internal) tissues we investigated (including the chalaza (see below)). The respective results further indicate that the final orientation of this polar axis relative to the long axis of the gynoecium is not established at the time the axis becomes apparent but is due to a more complex morphogenetic process involving a twist in the funiculus. This is also a new finding. We have reorganized the entire sections in the Results and the Discussion to clarify these important points.

The mechanism regulating early anterior-posterior polarity in the primordium is presently not known. Thus, we can only speculate. We provide some possible hints in the revised text (Discussion paragraph two).

The WUS expression pattern is interesting, but not connected at all to phenotypes of the wus mutant, hence it is unclear what lesson to take from this.

At one level, the analysis of *WUS* expression serves as proof-of-concept that the spatial control of gene activity can be investigated with single-cell resolution in 3D digital ovules. At another level there is exciting biology associated with *WUS* and ovule development. We rephrased much of this section to better introduce the known *WUS* function in the ovule (subsection “Scattered buildup of WUSCHEL expression in the nucellus”). Furthermore, we now outline several open questions relating to *WUS* expression in the ovule and provide a better summary regarding the relevance of our findings. Those questions are addressed in this study.

For example, data published by several labs indicate that *WUS* controls patterning in the chalaza and MMC formation. Moreover, recent evidence suggests that *WUS* expression has to be excluded from the MMC to allow regular meiosis. Available expression data suggested that *WUS* activity is restricted to the nucellar epidermis. These and other findings indicated that *WUS* functions in a non-cell-autonomous fashion in the ovule primordium. Interestingly, the *WUS* mechanisms in the ovule and shoot apical meristem (SAM) must differ as in the ovule there is no *CLAVATA* signaling and WUS does not appear to move between cells. It was unclear when *WUS* expression becomes detectable during ovule development and whether its expression is always restricted to the nucellus and the nucellar epidermis. We find *WUS* reporter signal only in the epidermis of the nucellus during stage 1-II. However, *WUS* expression is apparently not always restricted to the nucellar epidermis as we eventually observed reporter activity in L2 cells as well. Interestingly, though, we never detected reporter signal in the MMC (a L2 cell). Thus, our expression study generated new insight into the temporal and spatial establishment of *WUS* expression (strong temporal control of the initiation of *WUS* expression at early stage 1-II, initially scattered expression within the nucellar epidermis, eventually occasional signal in L2 cells of the nucellus). Our data also provide additional strong support for current ideas regarding *WUS* function in the ovule.

The continuous growth aspect is interesting, but it is not clear why this was tested. Was the hypothesis that there are growth pulses? Is the result unexpected?

Previously, it was unknown what type of temporal growth pattern underlies primordium outgrowth and thus it was unclear what to expect. We decided to address this issue and asked the question if ovule primordium outgrowth occurs in an even fashion or is subject to growth pulses. Our data suggest that the primordium undergoes continuous outgrowth. We now explicitly state in the introduction to this section that it was unknown what type of growth pattern happens during primordium formation (subsection “Even growth of ovule primordia”).

The interesting description of the ino mutant phenotype, and any discussion of the non-autonomous role in promoting inner cell growth and division, is only meaningful if INO protein accumulation is also shown in the same ovules, as was done for WUS.

We believe this comment can be separated into two parts. We beg to differ regarding the description of the *ino* phenotype. In our opinion that description stands for itself, irrespective of where *INO* is expressed. With the description of the *ino* phenotype we show what deep insights regarding gene function can be gained from a quantitative cellular analysis of a mutant 3D digital organ. Moreover, the analysis of *ino* reveals that this can be true even for a mutant phenotype that was already quite well studied using conventional approaches. The analysis revealed novel functions for *INO* in nucellar development and in the control of early ovule curvature (kink formation). In the revised manuscript, we better introduce *INO* in the Results (subsection “INNER NO OUTER affects development of the nucellus”). In addition, we emphasize more what can be learned from the *ino* mutant regarding ovule curvature in the Discussion (final paragraph).

We agree that a detailed cellular analysis of gene and protein expression is important to address the notion of a non-cell-autonomous function of *INO*. The current data in the literature are consistent and indicate that the presence of *INO* transcripts and INO protein is restricted to the epidermis and the outer layer of the outer integument (oi2). Nevertheless, this notion needs to be tested further taking advantage of the increased cellular resolution provided by 3D digital ovules. We rephrased the respective sections to better introduce the available *INO* expression data and emphasize the requirement for further studies.

The relevance of the identification of cell populations in the chalazal domain (Figure 12B/C) is not clear. Also, the definition of what cells belong to which domain seems fairly arbitrary. What criteria were used? There is a risk of circular argumentation here.

We reorganized this section and Figure 12 (new Figures 8 and 11). We came to realize that it made more sense to remove the characterization of the two different cell populations of the chalaza (now called anterior and posterior chalaza) from Figure 12 (control of kink formation) and put the results into a separate figure (new Figure 8). The new section covering Figures 7 and 8 provides a coherent anatomy-based outline of the establishment and maintenance of anterior-posterior polarity across ovule development (subsection “First morphological manifestation of polarity in the young ovule”). We now also provide a precise morphological definition of the two domains which is supported by data from a previous clonal analysis.

In sum, please make more of a clear case for how the pipeline and atlas can be used to discover new functions of known genes, or better explain a mutant phenotype.

We believe the revised manuscript addresses these issues.

Revisions expected in follow-up work:For example, localizing the INO protein in segmented ovules and/or a description of the wus phenotype using this pipeline would add more biological insight.

We agree with both issues. In fact, we are currently establishing lines that allow a quantitative assessment of *INO* expression in 3D digital ovules. They will be instrumental for our future work.

Studying *WUS* function in the ovule is not trivial. Apart from its function in ovule development *WUS* is essential for the development of shoot and floral meristems. As a consequence, *wus* mutants do not form pistils or ovules. Thus, to study the role of *WUS* in ovule development one has to restrict the manipulation of *WUS* activity to the ovule. Such lines were available in the past but have meanwhile been lost. As *WUS* plays a central role in early pattern formation during ovule development we plan to regenerate such lines and analyze the *wus* phenotype with the help of 3D digital ovules.